# Multi-Predict: Few Shot Predictors For Efficient Neural Architecture Search

Yash Akhauri[1]   Mohamed S. Abdelfattah[1]

[1]Cornell University

**Abstract**   Many hardware-aware neural architecture search (NAS) methods have been developed to optimize the topology of neural networks (NN) with the joint objectives of higher accuracy and lower latency. Recently, both accuracy and latency predictors have been used in NAS with great success, achieving high sample efficiency and accurate modeling of hardware (HW) device latency respectively. However, a new accuracy predictor needs to be trained for every new NAS search space or NN task, and a new latency predictor needs to be additionally trained for every new HW device. In this paper, we explore methods to enable multi-task, multi-search-space, and multi-HW adaptation of accuracy and latency predictors to reduce the cost of NAS. We introduce a novel search-space independent NN encoding based on zero-cost proxies that achieves sample-efficient prediction on multiple tasks and NAS search spaces, improving the end-to-end sample efficiency of latency and accuracy predictors by over an order of magnitude in multiple scenarios. For example, our NN encoding enables multi-search-space transfer of latency predictors from NASBench-201 to FBNet (and vice-versa) in under 85 HW measurements, a 400× improvement in sample efficiency compared to a recent meta-learning approach. Our method also improves the total sample efficiency of accuracy predictors by over an order of magnitude. Finally, we demonstrate the effectiveness of our method for multi-search-space and multi-task accuracy prediction on 28 NAS search spaces and tasks.

## 1 Introduction

Neural architecture search (NAS) has seen great success in discovering neural network (NN) architectures with high accuracy. However, NAS methods are often costly, requiring several hundreds of hours of compute. Recently, NAS methods have utilized surrogate models such as neural accuracy predictors to guide the search process [11, 21, 22]. The primary bottleneck with such methods is the training time needed to obtain a sufficient number of NN/accuracy samples used to train a predictor. Hardware-aware NAS additionally considers hardware (HW) latency as a search objective to find NNs that are both accurate and efficient [3, 6, 12]. In this case, a latency predictor is often used to estimate NN latency, thus incurring additional training time and on-device measurement samples [5, 8]. Both accuracy and latency predictors have demonstrated their effectiveness in the context of sample-based NAS, motivating further investigation of their efficiency, training method, and accuracy.

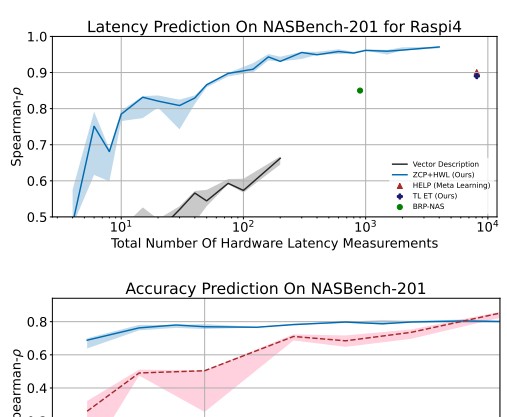

Figure 1: Effective NN encodings can improve the measurement & sample efficiency of predictors by over two orders of magnitude.

The key to efficient prediction-based NAS lies in minimizing the number of NN trained models for accuracy predictors, and HW measurements for latency predictors. In that vein, HELP [8] utilized metalearning to adapt a latency predictor to multiple devices.While this representative approach has begun to create generalizable/transferable predictors, it requires a very large number of NN accuracy or latency samples for *pretraining* transferable predictors. In this paper, we look broadly at the problem of generalizable but sample-efficient accuracy, and latency prediction. We make progress towards efficient multi-search-space, multi-task and multi-device deployment of NNs on the ever growing catalogue of HW platforms and novel NAS spaces. From Figure 1, we show that our methods improve the total sample efficiency of both latency and accuracy predictors by more than an order of magnitude. The main contributions of this paper are:

• **Few-Shot Accuracy And Latency Prediction Using Novel NN encodings**: We study two NN encodings (called ZCP and HWL) that enable accuracy and latency prediction in as few as 10 samples. Instead of representing NNs with a search-space dependent vector of its topology, ZCP and HWL represent NNs with a search-space independent vector of metrics based on zero-cost proxies or HW latency measurements respectively. Figure 2 shows an overview of our approach.

• **Multi-Device Latency Prediction Using Transfer Learning**: Our study shows that using learnable HW embeddings and a simple transfer learning strategy can double the efficiency of multi-device latency prediction compared to the latest metalearning approach [8].

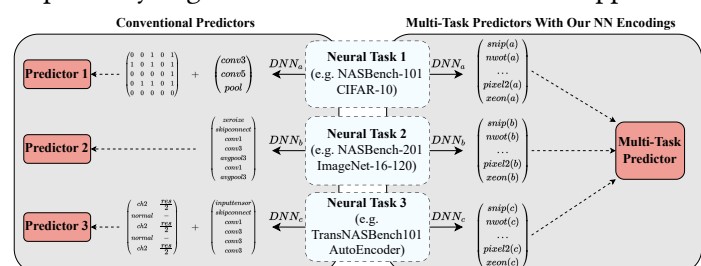

Figure 2: By representing NNs with a vector of their zero-cost proxies (ZCP) or HW latency measurements (HWL), we can use a single accuracy/latency predictor for multiple NAS search spaces. In contrast, we'd need to train multiple predictors for conventional vector (Vec) encodings.

• **Multi-Search-Space Latency & Accuracy Prediction**: We demonstrate transfer of HW latency predictors from one NAS search space to another in as few as 4 samples. Further, we demonstrate the efficacy of few-shot transfer of accuracy predictors across over 95 domain (search space + task) transfer pairs.

## 2  Related Work

**Prediction Based NAS**: Accuracy predictors are used to efficiently identify promising candidates from a NAS space as demonstrated by PNAS [11]. NPENAS [21] showed that accuracy predictors can be integrated with evolutionary search algorithm to perform NAS. Further, NPNAS [22] utilized an accuracy predictor to identify models with the top-K highest predicted accuracy, which were fully trained for evaluation. To improve the effectiveness of predictor-based NAS, BRP-NAS [5] introduced a predictor that learns a binary relation for accuracy prediction, integrating this with sample-based NAS delivered state of the art NAS results. A recent paper [16] explores transferring architectures from previously solved, related problems to treat NAS as a few-shot learning problem. Given the high sample efficiency of predictor-based NAS, we explore the transferability and generalizability of predictors for domain agnostic NAS with knowledge reuse.

**Latency modeling in NAS**: The accuracy of latency predictors for HW-aware NAS is prone to errors, and proxy metrics like FLOPs or model size have been used as alternatives. Previous layer-wise predictors [17] fail to account for the interactions between layers on actual HW. BRP-NAS [5] introduced an end-to-end NAS latency predictor based on a GCN. HELP [8] and MAPLE [14] utilize few-shot latency estimation methods to improve sample efficiency on their target HW, at the cost of expensive pre-training on existing HW measurements. Our aim is to create sample-efficient

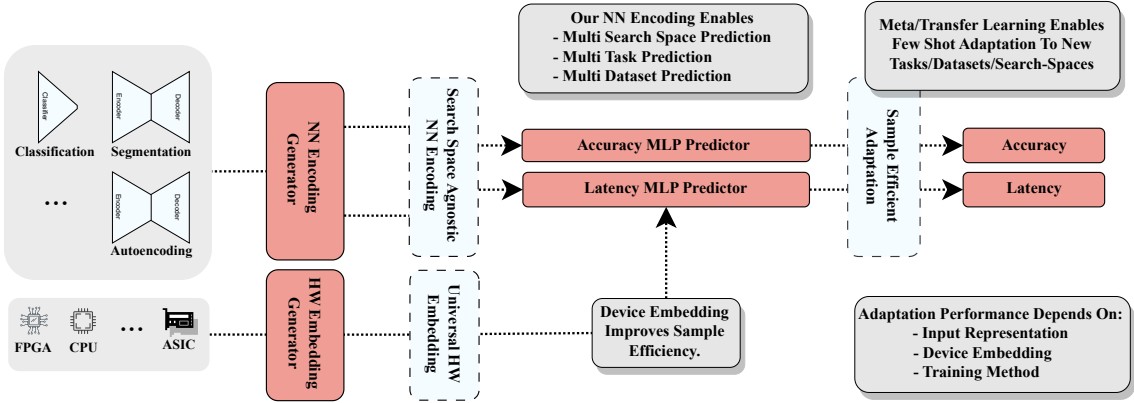

Figure 3: Predictors are used for both accuracy and latency to achieve SoTA results on sample-based NAS. In this work, we explore (a) New DNN encodings for prediction that enable multi-search-space NAS, (b) Transfer-learning methods to adapt a particular predictor to multiple tasks, devices, data-sets.

multi-HW latency predictors that require much fewer latency measurements for both pre-training, and on the target device.

**Zero Cost Proxies**: Zero Cost Proxies (ZCPs) [1] attempt to quantify the *trainability* and *expressivity* of NN architectures without training them. Several zero cost proxies are inspired by pruning at initialization research to generate scores for architectures [10, 13, 18, 19]. Several data-independent zero cost proxies have also been proposed [2]. NAS-Bench-Suite-Zero [7] evaluates 13 different zero cost proxies across 28 tasks and makes all of the results available for use. Prior work [1] has focused on using ZCPs as a weak predictor of accuracy within a NAS search. However, we utilize ZCPs in a novel way to represent NNs at the input of a predictor, thus enabling the transfer of predictors across domains more effectively.

## 3 Method

In this section, we describe our methodology of predicting the latency and accuracy of NN architectures. Specifically, we focus on generalizing the task of prediction such that it is agnostic to task, search-space and HW. Due to the domain agnostic encoding of NNs depicted in Figure 3, we can have one predictor for multiple NAS tasks, search spaces, and HW devices. We utilize a first order transfer learning (TL) strategy to train and transfer predictors. Note that we train separate predictors for latency and accuracy respectively. In doing so, we study **(1)** Learnable HW Embedding Tables for multi-device latency prediction and **(2)** Different DNN encoding formats (**ZCP**: Zero-Cost Proxies, **HWL**: HW Latencies) that improve predictor sample-efficiency and enable multi-task and multi-search-space adaptation.

The task of prediction in NAS can be generally defined by $\tau = \{\mathbf{X}^\tau, \mathbf{Y}^\tau\}$, where $\mathbf{X}^\tau \subset \mathcal{X}$ is a set of NN architectures and $\mathbf{Y}^\tau \subset \mathcal{Y}$ is the quantity to be predicted for $\mathbf{X}^\tau$, either accuracy $\mathbf{Y}_A^\tau$ or latency $\mathbf{Y}_L^\tau$. We train a four-layer MLP based regression model $f(x, \theta) : \mathcal{X} \to \mathbb{R}$, parametrized by $\theta$ by minimizing the empirical mean-squared differences (or loss $\mathcal{L}$) between the predicted values $f(\mathbf{X}^\tau; \theta)$ and the actual/measured values $\mathbf{Y}^\tau$ as shown here: $\min_\theta \mathcal{L}(f(\mathbf{X}^\tau; \theta), \mathbf{Y}^\tau)$. Typically, a different predictor needs to be trained for every different NAS search space, task, or HW device. In the following, we describe how a single predictor that can generalize to all of the above.

1. Expanding the latency prediction problem to *also* take the HW device $h \in \mathcal{H}$ as an input $f(x, h; \theta) : \mathcal{X} \times \mathcal{H} \to \mathbb{R}$. This allows us to predict latency for multiple devices using a single trained predictor. We primarily compare to HELP, who approached this problem with few-shot metalearning [8]. We show that HELP performs poorly when the *task distance* is high between

different devices, and we describe our simple transfer learning (TL) approach which when combined with new NN encodings and HW embeddings, performs better in many scenarios.

2. Investigating different input encodings for the NN ($x$). Most prior work has simply taken a *vector* encoding of the NN as input to the predictor. Instead, we experiment with novel NN encodings that do not reflect the NN topology, but rather a vector of measurements or computations (**R**) performed on the NN $\mathbf{R}(x) : \mathcal{X} \to \mathbb{R}^r$, where $r$ is the number of elements in the vector. We try two encodings: a vector of zero-cost proxies (**ZCP**), and a vector of HW latency measurements (**HWL**) on different devices. We will demonstrate that our new encodings improve prediction accuracy. More importantly, **R** is independent from the NN topology, and can therefore work for multiple search spaces, allowing us to train a single predictor for multiple NAS search spaces.

### 3.1 Hardware Embedding

For a latency predictor of the form $f(x, h; \theta)$ : $\mathcal{X} \times \mathcal{H} \to \mathbb{R}$, we need to find an appropriate method of representing $h \in \mathcal{H}$. We discuss three methods we explore for representing HW devices. **(1) Sample**: HELP [8] obtains the latencies of a set of fixed reference NN architectures to represent the HW device $h$. For instance, 10 fixed NN architectures are used to represent a hardware device in HELP. For each device, these 10 NNs are benchmarked on the device and their latencies are concatenated into a vector, which is the HW representation. Since HW devices may be diverse and heterogeneous, architectures used to generate this HW embedding may not be representative of the true characteristics of the device, and a larger set of NNs may be required to faith-

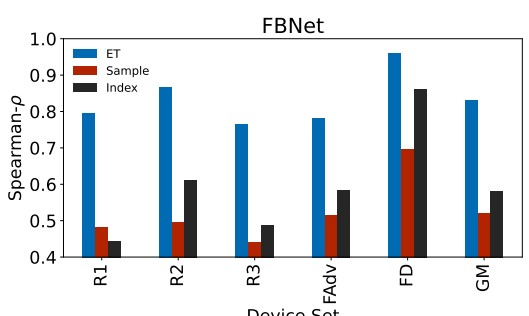

Figure 4: Learned HW Embedding Table improves latency prediction on FBNet over 5 train-test device sets. (R1–3: random subsets of FBNet, FAdv: FBNet-Adversarial, FD: FBNet-Default, GM: Geomean)

fully capture the device behavior. **(2) Index**: Instead of utilizing a set of latencies to represent the HW device, we can simply represent devices by the binary form of their index. However, this would mean that a new device would simply be represented by a binary embedding, which may be unrelated to the actual characteristics of the HW. **(3) Embedding Tables (ET)**: We initialize a *d-dimensional* learnable embedding table to represent the set of training devices. Each embedding lookup can be interpreted as using a one-hot vector $e_i$ to represent the $i^{th}$ device. Thus, obtaining the row vector corresponding to the $i^{th}$ HW device can be presented as $E_h = e_i E$, where $E \in \mathbb{R}^{|\mathcal{H}| \times d}$ is the embedding table. Figure 4 shows a preview of the results demonstrating that our proposed Embedding Tables outperforms the "Index" baseline and "Sample" based embeddings on a number of multi-HW latency prediction datasets.

### 3.2 NN Encoding

Conventional **Vec** (Vector) encoding of NN structure typically consists of an adjacency matrix to describe connectivity and a list of operations [23]. However, **Vec** encoding ($\mathbf{X}^\tau$) for different search spaces may be vastly different in dimensionality and semantics, making it *impossible* to create transferable predictors across search spaces as shown in Figure 2. Instead, we propose to use a vector of NN metrics as an encoding of a NN for training a predictor. Several metrics can be generated for any NN, irrespective of task and search-space. For instance, the HW latency can be measured for any NN in a reliable fashion. Further, we can also generate scores for NNs by utilizing Zero-Cost Proxies [1]. These metrics can be concatenated to generate a continuous vector encoding for NN architectures. We thus present two novel forms of neural architecture encodings:

- **HW Latencies (HWL)**: Suppose we have $l$ devices available for measurement of latency of arbitrary NN architectures. We introduce a function $\mathbf{L}(x^\tau) : \mathcal{X} \to \mathbb{R}^l$, which maps any NN architecture to a real tensor of $l$ elements. Each of the elements of the tensor correspond to the normalized latency of the architecture on the $l^{th}$ device.
- **Zero Cost Proxies (ZCP)**: Suppose we have $z$ zero cost proxies available for generating scores of arbitrary NN architectures. We introduce a function $\mathbf{Z}(x^\tau) : \mathcal{X} \to \mathbb{R}^z$, which maps any NN architecture to a real tensor of $z$ elements. Each of the elements of the tensor correspond to the normalized score of the architecture on the $z^{th}$ ZCP.

### 3.3 Few-Shot Adaptation

We describe our method of few-shot adaptation of a trained MLP predictor $f$ with the latency measurements or accuracy samples collected from the target HW device or neural task respectively.

**Hardware Adaptation**: We wish to train on a set of HW devices denoted by $\tau^1$. For prediction on a novel HW device denoted by $\tau^2 = \{h^{\tau^2}, \mathbf{X}^{\tau^2}, \mathbf{Y}^{\tau^2}\}$, we have $|\tau^2|$ sample measurements and $|\tau^2| \ll |\tau^1|$. For embedding tables, we initialize a new row in our table for a new HW device. The values of this row are assigned to be the same as a device $h^{\tau^1}$ such that the correlation between the device $h^{\tau^2}$ and $h^{\tau^1}$ is maximized for a small set of sampled NN latencies $\{\tilde{\mathbf{Y}}^{\tau^1}, \tilde{\mathbf{Y}}^{\tau^2}\}$ as shown in Eq. 1. Our method uses the same NNs $\tilde{\mathbf{X}}^{\tau^2}$ for embedding initialization as it does for fine-tuning.

$$E(h_{\tau^2}) = E(\underset{h \in \mathcal{H}}{\arg\max} \, \rho(\tilde{\mathbf{Y}}^{\tau^1}, \tilde{\mathbf{Y}}^{\tau^2})) \tag{1}$$

For index/sample HW embeddings, we either assign the next available index or collect latency samples to generate the appropriate embedding for the HW device $h^{\tau^2}$. After initializing the embedding function $E$, we perform simple first order fine-tuning of the NN as shown in Eq. 2, where $\{\tilde{\mathbf{X}}^\tau, \tilde{\mathbf{Y}}^\tau\}$ refers to a training set.

$$\underset{\theta}{\min} \, \mathcal{L}(f(\tilde{\mathbf{X}}^{\tau^2}, E(h^{\tau^2}); \theta), \tilde{\mathbf{Y}}^{\tau^2}) \tag{2}$$

**Search Space Adaptation** When adapting a latency or accuracy prediction model from one search space to another, it is likely that $|x^{\tau^1}| \neq |x^{\tau^2}|$. This is depicted in Figure 2, every search space has its own encoding. As a mapping does not exist between domains, we utilize the ZCP $\mathbf{Z}$ and HWL $\mathbf{L}$ embeddings to represent NN architectures across multiple search spaces. Thus, the model that we train and fine-tune is $f(\mathbf{R}(x); \theta) : \mathbb{R}^r \to \mathbb{R}$. Here, $\mathbf{R} \subseteq \{\mathbf{L}(x), \mathbf{Z}(x)\}$. The functions $\mathbf{L}$ and $\mathbf{Z}$ are generated by executing NNs on reference devices or calculating a set of zero cost proxies respectively. The generated latencies and scores are search-space independent since $|R(x^{\tau^1})| = |R(x^{\tau^2})|$, allowing transfer from one neural search space to another using the same predictor $f$. Therefore a pre-trained predictor $f(\mathbf{R}(\mathbf{X}^{\tau^1}); \theta) : \mathbb{R}^r \to \mathbb{R}$ can be fine-tuned on a target task $\tau^2 = \{\mathbf{X}^{\tau^2}, \mathbf{Y}^{\tau^2}\}$ by minimizing empirical loss on a training set $\{\tilde{\mathbf{X}}^{\tau^2}, \tilde{\mathbf{Y}}^{\tau^2}\}$:

$$\underset{\theta}{\min} \, \mathcal{L}(f(\mathbf{R}(\tilde{\mathbf{X}}^{\tau^2}); \theta), \tilde{\mathbf{Y}}^{\tau^2}) \tag{3}$$

## 4 Hardware Latency Predictors

In this section, we look at the task of HW latency prediction. We verify the efficacy of our training and fine-tuning method for few-shot learning of latency predictors on novel HW devices as well as empirically assess the effectiveness of different methods of encoding these devices. Finally, we utilize our proposed NN encodings (**ZCP** and **HWL**) to transfer knowledge from the FBNet [24] search space to the NASBench-201 search space and vice versa for latency prediction. Previous work has defined sample efficiency as the amount of new data required when transferring a pre-trained predictor to a different hardware device [8, 14]. Although we acknowledge that optimizing for the number of new samples required for transfer is crucial, we also believe that the number

|  | NASBench-201 Default Task | | | | | | | |
|---|---|---|---|---|---|---|---|---|
| Method | Samples | GPU | CPU | Pixel2 | Raspi4 | ASIC | FPGA | Mean |
| FLOPs | - | 0.95 | 0.83 | 0.77 | 0.85 | 0.44 | 0.9 | 0.79 |
| BRP-NAS | 900 | 0.81 | 0.8 | 0.67 | 0.85 | 0.81 | 0.8 | 0.79 |
| BRP-NAS (+ES) | 3200 | 0.82 | 0.81 | 0.69 | 0.85 | 0.83 | 0.83 | 0.81 |
| HELP | 20* | 0.98 | 0.99 | 0.8 | 0.89 | 0.94 | 0.99 | 0.93 |
| TL Vec Sample | 20* | 0.87 | 0.92 | 0.94 | 0.87 | 0.82 | 0.69 | 0.85 |
| TL Vec Index | 10 | 0.94 | 0.93 | 0.87 | 0.77 | 0.73 | 0.9 | 0.86 |
| TL Vec ET | 10 | 0.99 | 0.95 | 0.88 | 0.9 | 0.9 | 0.99 | 0.94 |
| **TL ZCPVec ET** | **10** | **0.97** | **0.96** | **0.86** | **0.91** | **0.95** | **0.97** | **0.94** |
| TL HWLVec ET | 10 | 0.95 | 0.97 | 0.81 | 0.88 | 0.93 | 0.95 | 0.91 |

*20 samples are required, with 10 dedicated to creating the "sample" HW embedding.

Table 1: Spearman-$\rho$ of TL and existing methods on NASBench201 for Latency Prediction. (ES: Extra Samples)

| NB201-Adversarial Task | | |
|---|---|---|
| Method | Samples | Mean |
| HELP | 20 | 0.36 |
| TL Vec ET | 20 | 0.65 |
| **TL ZCPVec ET** | **20** | **0.78** |
| TL HWLVec ET | 20 | 0.73 |

| FBNet-Adversarial Task | | |
|---|---|---|
| Method | Samples | Mean |
| HELP | 20 | 0.37 |
| TL Vec ET | 20 | 0.39 |
| **TL ZCPVec ET** | **20** | **0.45** |
| TL HWLVec ET | 20 | 0.41 |

Table 2: Spearman-$\rho$ of adversarial device sets.

| FBNet Default Task | | | | |
|---|---|---|---|---|
| Method | Samples | FPGA | Raspi | ASIC | Mean |
| HELP | 20 | 0.89 | 0.94 | 0.89 | 0.91 |
| TL Sample | 20 | 0.74 | 0.79 | 0.81 | 0.78 |
| TL Index | 10 | 0.92 | 0.86 | 0.87 | 0.88 |
| **TL ET** | **10** | **0.96** | **0.95** | **0.98** | **0.96** |

Table 3: Spearman-$\rho$ of TL and existing methods on FBNet for Latency Prediction.

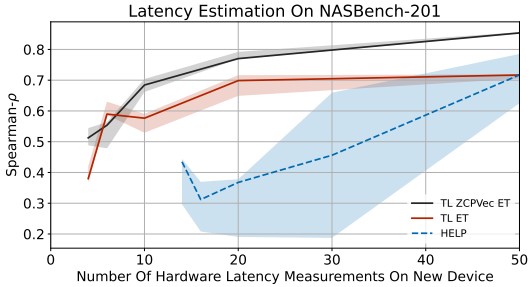

Figure 5: ZCPs with Vec encoding improves latency prediction on NB201-Adversarial Task.

of pretraining samples should be taken into account. It is important to note that more efficient predictors will require fewer pretraining samples.

**Few-Shot Latency Predictor *Transfer*:** The task of learning a reliable latency predictor for a neural architecture search space on a specific HW requires a large number of samples to avoid overfitting [8]. This has to be repeated for every HW and search space. HELP [8] looks at the task of using meta-learning to transfer a single latency predictor to multiple target devices and architectures from an unseen space. It utilizes 900/4000 latency samples for a set of training devices on the NASBench-201/FBNet spaces respectively to train a baseline predictor. The predictor is then tested on the remaining 14725/1000 latency points for the hardware on NASBench-201/FBNet spaces respectively. Then, this predictor is transferred to predict latency on a target (test) HW with only 20 samples. It performs extremely well on the reported train-test device sets (referred to as *Default Tasks*) on the NASBench-201 and FBNet NAS spaces. We find that one of the key reasons for this is the low task distance (high correlation) between the training and test device sets. Task distance refers to the latency or accuracy correlation between the training and test device/NAS space respectively. We conduct a deeper investigation of task distance and its effect on latency predictors in the Appendix. Further, we introduce *adversarial* device sets, named 'NB201/FBNet-Adversarial Task' for the NASBench-201 and FBNet NAS space, which exhibits low train-test device correlation.

From Table 1 and Table 3, we can see that our method is able to perform on par or better than HELP [8] with half as many samples—this is on the "default" hardware set defined in the HELP paper [8]. One of the reasons why the sample efficiency is doubled is due to the embedding table that is utilized to represent the HW which improves Spearman-$\rho$ from 0.85 to 0.94 in Table 1.

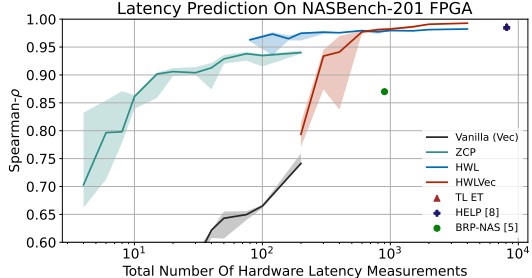

Figure 6: Training from scratch with ZCP/HWL encodings achieves a similar Spearman-$\rho$ to HELP in under 80 **total** measurements, a 100× improvement.

Table 4: Reducing number of samples from pre-training device set can negatively impact prediction spearman-$\rho$.

| FBNet Default Task | | |
|---|---|---|
| Pre-Train Samples | HELP | TL ET |
| 4000 | 0.91 | 0.96 |
| 1000 | 0.85 | 0.91 |
| NASBench201 Default Task | | |
| Pre-Train Samples | HELP | TL ET |
| 900 | 0.93 | 0.94 |
| 50 | 0.91 | 0.91 |

Figure 4 shows consistent benefit of using a learned device embedding on the FBNet space on the adversarial task. Table 2 compares our method to HELP on a new Adversarial Task. Since the default hardware set had a high input output correlation, we define an adversarial task as one where the test devices have low spearman-$\rho$ correlation between train and test devices. We use 20 samples for tests here to emphasize on the importance of the ZCP and HWL representation in improving spearman-$\rho$. For this task, the task distance between the pretraining set and the test set is much higher than the default set. In this more challenging task, we observe a much higher improvement in Spearman-$\rho$ using TL ZCPVec compared to HELP for both NASBench-201 and FBNet search spaces, demonstrating the superiority of our approach on a more challenging task. Figure 5 plots more details for the NASBench-201 Adversarial Task, showing that HELP requires a much higher number of samples to catch up to our TL methods, especially when ZCP is used.

**Few-Shot Latency Predictor *Pre-training***: While Transfer Learning and HELP [8] are effective ways of transferring latency predictors while minimizing the number of samples required on the new device, they need to first train a predictor on a large set of training devices. As discussed in the previous section, the effectiveness of such methods highly depend upon the task distance (train-test device correlation). Further, to train the predictor, 900 and 4000 latency samples are required for 18 devices on NASBench-201 and FBNet respectively. Table 4 shows that the prediction accuracy decreases when the number of pretrain samples are decreased, however, our *TL ET* method is impacted less than the HELP baseline. In general, the number of *total* HW latency measurements of these methods are very high, as several training devices are required to minimize the task distance when adding a new hardware device.

The *TL ZCPVec ET* method depicted in Figure 5 and Table 1 inputs the Vec (vector) description of an architecture along with its ZCP description. This simple change significantly improves the accuracy of the latency predictor without increasing the number of HW latency measurements required. We go one step further and look at training latency predictors **from scratch** using the ZCP and HWL encodings. In Figure 6, we train latency predictors from scratch with the Vec, ZCP, HWL and HWLVec encodings. We find that the total number (both pretraining and target device) of HW latency measurements required for latency predictors with the ZCP and HWL encoding is significantly lower compared to methods like BRP-NAS [5], HELP [8], and our TL variants. Thus, we find that ZCP and HWL are not only a search space agnostic encoding of NNs, but also ones that can do effective few-shot latency prediction from scratch.

**Multi-Search-Space Latency Prediction**: A search-space agnostic encoding of NNs akin to the one depicted in Figure 2, can not only enable few-shot training of predictors from scratch, but also enable transfer of a latency predictor from one neural search space to another. Since the vector (Vec) encoding of candidate architectures from the FBNet space is different from that of the NASBench-201 space, it is not possible to transfer knowledge from one predictor to another.

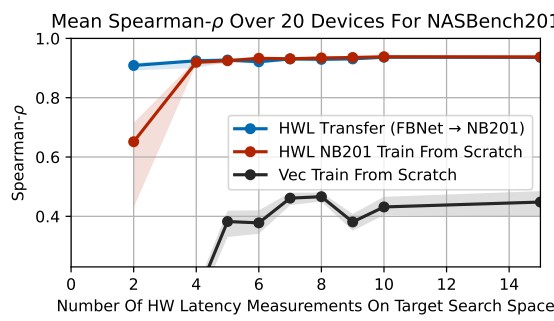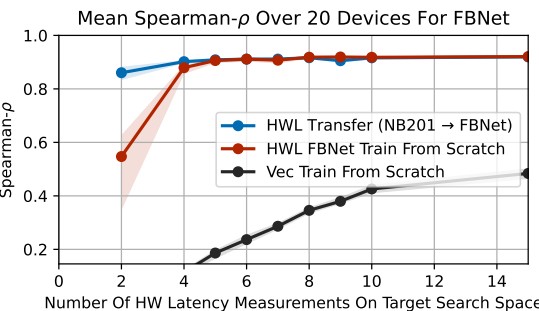

Figure 7: Using the HWL encoding to transfer a predictor from a source search space to a target search space demonstrates superior performance than training from scratch.

However, the HWL encoding is independent from the NN search-space. We thus utilize the HWL encoding to train on 15% of the architectures on one space, and then transfer to the other space with very few samples (X-axis of Figure 7). Note that this transfer across search spaces is different from the transfer across hardware devices presented in HELP [8] and in our previous sections. Figure 7 reinforces our results from the previous section, that training from scratch with the HWL requires far fewer samples than Vec to train an accurate predictor. Furthermore, when transferring a trained predictor from one search space to another, HWL proves to be advantageous. We believe that this is first time that knowledge from one NAS search space was utilized for a different NAS search space for latency prediction—our results look promising. In the Appendix, we demonstrate predictor transfer for 21 devices, from FBNet to NASBench-201 and vice versa.

## 5 Accuracy Predictors

In this section, we study few-shot accuracy prediction. First, we study the impact of our proposed NN encodings (ZCP and HWL) on training accuracy predictors from scratch. We then investigate the transfer of accuracy predictors across multiple search-spaces and multiple tasks in TransNASBench-101, and across both the Micro and Macro search spaces that are available in the benchmark [4]. Finally, we transfer an accuracy predictor from NASBench-201 to NASBench-301 [26] and vice versa for further empirical results.

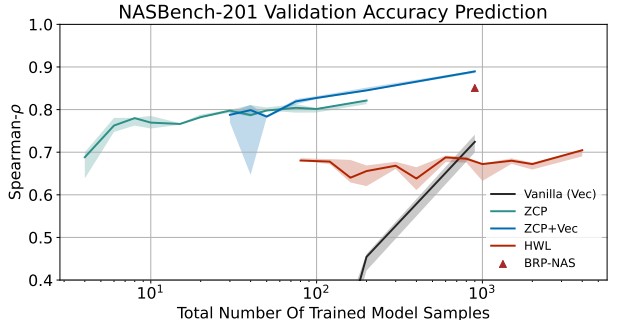

Figure 8: ZCP is more measurement efficient than other encodings for accuracy prediction.

**Few Shot Accuracy Prediction from Scratch**: We train a predictor from scratch on the NASBench-201 space to predict the validation accuracy on the CIFAR-10 data-set. Figure 8 depicts the trained model sample efficiency of the ZCP encoding with respect to other NN encodings. We find that ZCP is able to approximately match the Spearman-$\rho$ of BRP-NAS with less than half the samples. To understand why ZCP works better than the simple Vec encoding, we study the effect of incrementally removing zero cost proxies from the encoding in the Appendix. We show that prediction accuracy is resilient to the removal of multiple ZCPs from our NN encoding.

**Multi Search-Space and Task Accuracy Prediction**: We train individual predictors from scratch on every task from TransNASBench-101 Micro search space using 15% of the architectures, and transfer it to every other task on the TransNASBench-101 Micro and Macro search space with only 10 samples. This transfer task uses the ZCP NN encoding, and is compared with a train from

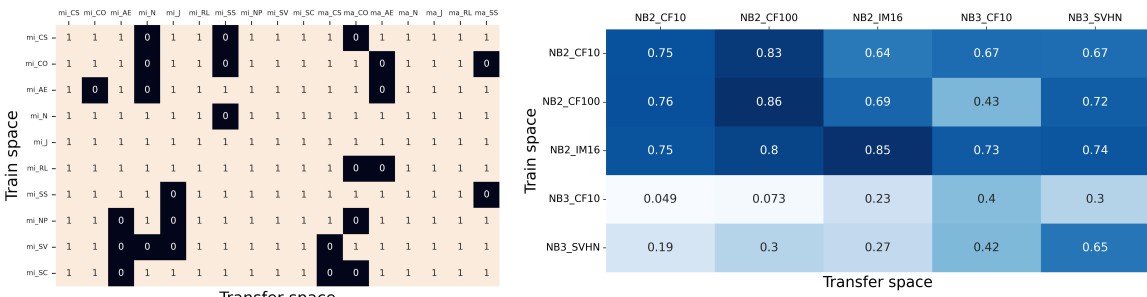

Figure 9: We train accuracy predictors on the 'Train Space' and transfer it to the 'Transfer Space' with the ZCP NN encoding (ZCP Transfer), and compare it with Vec train-from-scratch. (Left) 1 indicates an improvement in accuracy prediction. (Right) Numbers represent the *improvement* in accuracy prediction. Improvement is the difference in spearman-$\rho$ for the predictors.

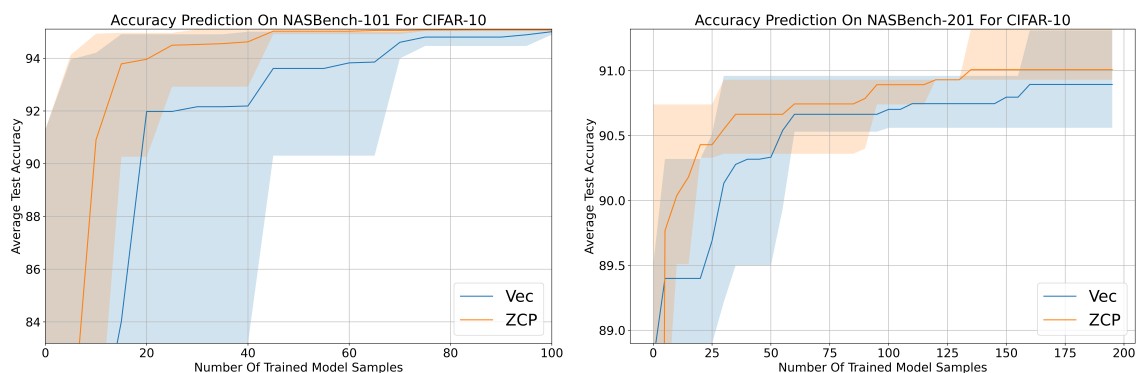

Figure 10: We compare the sample efficiency of Vec and ZCP in neural architecture search.

scratch strategy with the Vec NN encoding on every TransNASBench-101 Micro and Macro space. From Figure 9 (left), we see a benefit of multi-search-space and multi-task transfer learning in over 85% of the cases. We also do the same for three data-sets of NASBench201 and two data-sets of NASBench301, and find a consistent improvement in Spearman-$\rho$ for ZCP with transfer learning in Figure 9 (right). We have demonstrated effective transfer of accuracy predictors across tasks and search-spaces, utilizing previously learned knowledge to drastically reduce the number of samples on the new task or search space while considerably outperforming conventional train-from-scratch predictors.

**NAS Search Effectiveness**: To assess the efficacy of accuracy predictors utilizing ZCP input representation, we develop a straightforward search algorithm akin to [5]. For the NASBench-101 and NASBench-201 search spaces, we generate ZCP representations for all 423k and 15k neural network architectures, respectively. At each step, we sample 10 neural network architectures from the search space and train the accuracy predictor using these points. Subsequently, the accuracy predictor is employed to predict the accuracy of every neural network in the search space, and the top 10 points are selected as new samples. Accuracies of previously sampled neural network architectures are excluded when sampling new points. From Figure 10, we see that on the NASBench-101 search space, ZCP finds a near-optimal architecture in 40 samples, whereas Vec representation requires over 100 samples. A similar improvement in sample efficiency is observed for NASBench-201.

## 6 Discussion and Conclusion

In this paper, we introduced **Multi-Predict**, a first step towards search-space, task and device agnostic predictors for neural architecture search.

We studied two NN encodings (ZCP and HWL) that can enable **few-shot transfer** of knowledge from one search-space to another for the first time. We performed extensive experiments on many NAS search spaces, tasks, and devices that consistently show significant improvements in sample efficiency compared to training a conventional accuracy/latency predictor from scratch. We hope that our work can be a first step in developing NAS methods for much larger and more open-ended search spaces. Furthermore, such NN encodings enable NAS predictors that can continuously learn from prior NAS runs, even on different search spaces or tasks. While our initial results seem very promising, we believe that a more rigorous and extensive evaluation of these new NN encodings is warranted, for example, our investigation in the Appendix has already identified a failure scenario, when our ZCP NN encoding loses the most highly-correlated proxies. Developing more robust search-space independent NN encodings, and evaluating their encoding ability in a principled way is therefore a key challenge moving forward to enable our vision of extending NAS beyond a single search space.

Our paper has also identified methods to more accurately train HW latency predictors for multiple devices, including learnable HW embeddings that outperform previous metalearning approaches [8]. We made a key observation, that *task distance* or device latency correlation in this case, plays a large role in enabling such multi-device predictors. Further, we showed that our first-order transfer learning method with our HW embedding performs better than prior work in adversarial scenarios when the task distance is high.

The objective of our paper was to enable re-use of knowledge across existing HW latency and accuracy samples for a generalizable, few-shot approach to prediction-based NAS. We obtain over an order of magnitude improvement in the sample and measurement efficiency of latency and accuracy prediction. We also demonstrated multi-search-space and multi-task transfer of accuracy predictors over 28 NAS search spaces and tasks. Further, we enabled multi-search-space adaptation of HW latency predictors in under 5 samples. In the future, we intend to conduct deeper investigations of search-space agnostic encodings for NN and HW, and new training techniques to improve the sample efficiency and generalizability of predictors for NAS.

## 7 Broader Impact Statement

The pursuit of better representations for neural architecture search (NAS) can yield more efficient and high-performing neural network designs while also reducing the carbon footprint. Our research emphasizes the effectiveness of two computationally efficient methodologies for representing neural architectures, which represents a significant step towards achieving sample-efficient NAS. The HWL representation requires dedicated hardware infrastructure, but its sample efficiency through hardware inference would greatly reduce the number of architectures that require training. We are optimistic that this and other efforts to efficiently represent neural architectures can have a positive environmental impact by decreasing the cost of NAS.

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

## A Supplementary Materials for Multi-Predict: Few Shot Predictors For Efficient Neural Architecture Search.

### A.1 Neural Architecture Search Spaces

In this paper, we utilize several different neural architecture design spaces. NASBench-101 and NASBench-201 are cell based search spaces, consisting of 423,624 and 15,625 architectures respectively. NASBench-101 is trained on CIFAR-10, and NASBench-201 is trained on CIFAR-10, CIFAR-100 and ImageNet16-120. NASBench-301 is a surrogate NAS benchmark with $10^{18}$ total architectures. TransNAS-Bench-101 is a NAS benchmark with a *micro* (cell based) search space with 4096 architectures, and a *macro* search space with 3256 architectures. Each of these networks are trained on seven tasks from the Taskonomy data-set. These search spaces are unified within the NASLib framework. This space has further been extended by NAS-Bench-Suite-Zero, adding two data-sets from NAS-Bench-360, SVHN and four data-sets from Taskonomy. FBNet constructs a layer-wise search space which is more hardware-friendly than NAS-Bench-201, with $10^{21}$ unique architectures.

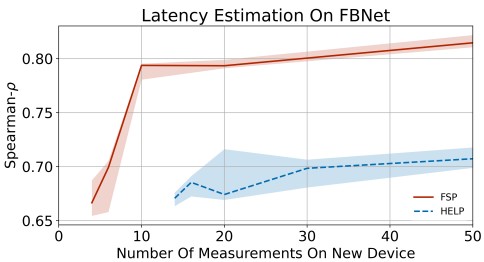

Figure 11: Sample efficiency of Transfer Learning (FSP) with respect to HELP on the FBNet search space for the FBNet-Adversarial Task.

| Name | Type |
|---|---|
| fisher [19] | Pruning-at-init |
| flops [15] | Baseline |
| grad-norm [1] | Pruning-at-init |
| grasp [20] | Pruning-at-init |
| l2-norm [1] | Baseline |
| jacov [13] | Jacobian |
| nwot [13] | Jacobian |
| params [15] | Baseline |
| plain [1] | Baseline |
| snip | Pruning-at-init |
| synflow [18] | Pruning-at-init |
| zen-score [10] | Piece. Lin. |

Table 6: List of ZC proxies in NAS-Bench-Suite-Zero [7]. These are used in our ZCP description.

### A.2 Hardware Latency Benchmarks

One of the key challenges of hardware aware neural architecture search is the collection of reliable hardware resource metrics for neural networks on the target search space. Several methods utilize simple metrics such as FLOPs as a metric for resource aware NAS, but it has been shown that design spaces with comparable FLOPs can have vastly different behavior on hardware [9]. HW-NAS-Bench introduces a public hardware latency data-set of two SoTA NAS search spaces (NAS-Bench-201 and FBNet). HW-NAS-Bench provides the measured/estimated hardware cost on six devices for all

Table 5: NAS Search Spaces and their respective metrics utilized in our paper.

| Search Space | HW Latency (HWL) | ZC Proxies (ZCP) | Num Architectures |
|---|---|---|---|
| NASBench101 [25] | ✗ | ✓ | 10000 |
| NASBench201 | ✓ | ✓ | 15625 |
| NASBench301 [26] | ✗ | ✓ | 11221 |
| TransNASBench101 Micro [4] | ✗ | ✓ | 3256 |
| TransNASBench101 Macro [4] | ✗ | ✓ | 4096 |
| Additional (NASBenchSuiteZero) [7] | ✗ | ✓ | 600 |
| FBNet [24] | ✓ | ✗ | 5000 |

| Device | Type | NASBench-201 | FBNet |
|---|---|---|---|
| HELP & HW-NAS-Bench [8, 9] | | | |
| 1080ti_1 | GPU | ✓ | ✓ |
| 2080ti_1 | GPU | ✓ | ✓ |
| 1080ti_32 | GPU | ✓ | ✓ |
| 2080ti_32 | GPU | ✓ | ✓ |
| 1080ti_256 | GPU | ✓ | ✓ |
| 2080ti_256 | GPU | ✓ | ✓ |
| titan_rtx_1 | GPU | ✓ | ✓ |
| titanx_1 | GPU | ✓ | ✓ |
| titanxp_1 | GPU | ✓ | ✓ |
| titan_rtx_32 | GPU | ✓ | ✓ |
| titanx_32 | GPU | ✓ | ✓ |
| titanxp_32 | GPU | ✓ | ✓ |
| titan_rtx_256 | GPU | ✓ | ✓ |
| titanx_256 | GPU | ✓ | ✓ |
| titanxp_256 | GPU | ✓ | ✓ |
| gold_6240 | CPU | ✓ | ✓ |
| silver_4114 | CPU | ✓ | ✓ |
| silver_4210r | CPU | ✓ | ✓ |
| gold_6226 | CPU | ✓ | ✓ |
| samsung_a50 | Mobile | ✓ | ✓ |
| pixel3 | Mobile | ✓ | ✓ |
| samsung_s7 | Mobile | ✓ | ✓ |
| essential_ph_1 | Mobile | ✓ | ✓ |
| pixel2 | Mobile | ✓ | ✓ |
| fpga | FPGA | ✓ | ✓ |
| raspi4 | RasPi | ✓ | ✓ |
| eyeriss | ASIC | ✓ | ✓ |
| EAGLE[5] | | | |
| core_i7_7820x_fp32 | Desktop CPU | ✓ | ✗ |
| snapdragon_675_kryo_460_int8 | Mobile CPU | ✓ | ✗ |
| snapdragon_855_kryo_485_int8 | Mobile CPU | ✓ | ✗ |
| snapdragon_450_cortex_a53_int8 | Mobile CPU | ✓ | ✗ |
| edge_tpu_int8 | Embedded TPU | ✓ | ✗ |
| gtx_1080ti_fp32 | Desktop GPU | ✓ | ✗ |
| jetson_nano_fp16 | Embedded GPU | ✓ | ✗ |
| jetson_nano_fp32 | Embedded GPU | ✓ | ✗ |
| snapdragon_855_adreno_640_int8 | Mobile GPU | ✓ | ✗ |
| snapdragon_450_adreno_506_int8 | Mobile GPU | ✓ | ✗ |
| snapdragon_675_adreno_612_int8 | Mobile GPU | ✓ | ✗ |
| snapdragon_675_hexagon_685_int8 | Mobile DSP | ✓ | ✗ |
| snapdragon_855_hexagon_690_int8 | Mobile DSP | ✓ | ✗ |

Table 7: Device List and their availability for NASBench-201 and FBNet

| Name | Type |
|---|---|
| NASBENCH101 CIFAR10 | NB1_CF10 |
| NASBENCH201 CIFAR10 | NB2_CF10 |
| NASBENCH201 CIFAR100 | NB2_CF100 |
| NASBENCH201 IMAGENET16-120 | NB2_IM16 |
| NASBENCH201 NINAPRO | NB2_NP |
| NASBENCH201 SVHN | NB2_SVHN |
| NASBENCH201 SCIFAR100 | NB2_SC100 |
| NASBENCH301 CIFAR10 | NB3_CF10 |
| NASBENCH301 NINAPRO | NB3_NP |
| NASBENCH301 SVHN | NB3_SVHN |
| NASBENCH301 SCIFAR100 | NB3_SC100 |
| TRANSBENCH101 MACRO CLASS SCENE | ma_CS |
| TRANSBENCH101 MACRO CLASS OBJECT | ma_CO |
| TRANSBENCH101 MACRO AUTOENCODER | ma_AE |
| TRANSBENCH101 MACRO NORMAL | ma_N |
| TRANSBENCH101 MACRO JIGSAW | ma_J |
| TRANSBENCH101 MACRO ROOM LAYOUT | ma_RL |
| TRANSBENCH101 MACRO SEGMENTSEMANTIC | ma_SS |
| TRANSBENCH101 MICRO CLASS SCENE | mi_CS |
| TRANSBENCH101 MICRO CLASS OBJECT | mi_CO |
| TRANSBENCH101 MICRO AUTOENCODER | mi_AE |
| TRANSBENCH101 MICRO NORMAL | mi_N |
| TRANSBENCH101 MICRO JIGSAW | mi_J |
| TRANSBENCH101 MICRO ROOM LAYOUT | mi_RL |
| TRANSBENCH101 MICRO SEGMENTSEMANTIC | mi_SS |
| TRANSBENCH101 MICRO NINAPRO | mi_NP |
| TRANSBENCH101 MICRO SVHN | mi_SV |
| TRANSBENCH101 MICRO SCIFAR100 | mi_SC |

Table 8: Short-hand names for NAS Search Spaces used in the correlation diagrams.

46875 architectures on NAS-Bench-201 across CIFAR-10, CIFAR-100 and ImageNet16-120. Further, the measured/estimated hardware-cost on these devices is also provided for all $10^{21}$ architectures in the FBNet search space. HELP [8] considers a hardware latency data-set of 7 representative platforms on the NAS-Bench-201, FBNet and MobileNetV3 search spaces. BRP-NAS [5] further provides LatBench/Eagle, a latency data-set for NAS-Bench-201 on six devices.

In Table 5, we summarize the search spaces we used in our paper. Further, in Table 7, we detail the hardware platform we used in our paper for analysis, as well as which search spaces their latency is available on.

### A.3 Zero Cost Proxies

We utilize the zero cost proxies listed in Table 6. The correlations between different Zero Cost Proxies *for the NASBench-201 CIFAR-10 search space* are provided in Figure 13 (a).

### A.4 Hardware Devices And Correlations

We present the correlation between hardware device latencies on the NASBench-201 search space with respect to ZC Proxies in Figure 13 (b). Further, in Figure 14, we present device correlations in three categories. 1 represents the device correlation is greater than 0.7, 0.5 indicates that the device correlation is greater than 0.5, but lesser than 0.7, finally, 0 indicates that the device correlation is less than 0.5. We find that most devices have a high correlation alternative. This indicates that

our HWL encoding can be an effective tool to map to arbitrary hardware platforms with high confidence.

## A.5 Importance of train-test device correlation

Meta-Learning performs extremely well when the task distance is low. One of the reasons for the success of HELP [8] in building accurate latency predictors is the low task distance. A low task distance can be established by measuring the Spearman-$\rho$ between the training and test devices. We take the default FBNet and NASBench-201 device sets reported by HELP and study their task distance. Table 11 indicates that in most cases, simply choosing a highly correlated device would perform better than utilizing transfer learning. However, it is important to note that establishing correlation between devices can be a tricky task, and a higher variance may be observed with different architecture samples. From Table 12, we find that if we remove devices with high train-test correlation from the training set, the performance of transfer learning falls extremely fast. Thus, while HELP and Transfer Learning are sample efficient methods of building hardware latency predictors, there exists a trade-off between training device correlation and sample efficiency. The higher the task distance, the more samples are likely to be needed to build a robust latency predictor.

## A.6 True Sample Efficiency Of HELP and BRP-NAS

In Appendix A.5, we discuss that Table 11 demonstrates a Spearman rank correlation between 0.83 and 0.97 for the training devices of the FPGA device. This indicates that scaling the total number of hardware measurements is not an efficient method for evaluating sample efficiency on the default device set. To augment Figure 6, we create an adversarial device set with train-test device correlations ranging from 0.5 to 0.7 for HELP. From Figure 16, it is evident that ZCP significantly enhances the total number of hardware latency measurements. It is worth mentioning that our HWL and HWLVec perform exceptionally well due to high train-test device correlations, akin to those observed for HELP in Figure 6.

## A.7 Multi-Search-Space Latency Predictor Transfer

In Figure 20 and 21, we present the per-device result of transferring a latency predictor from FBNet to NASBench201 and NASBench201 to FBNet respectively. In almost all cases, there is a benefit to HWL Transfer in the extremely low sample regime.

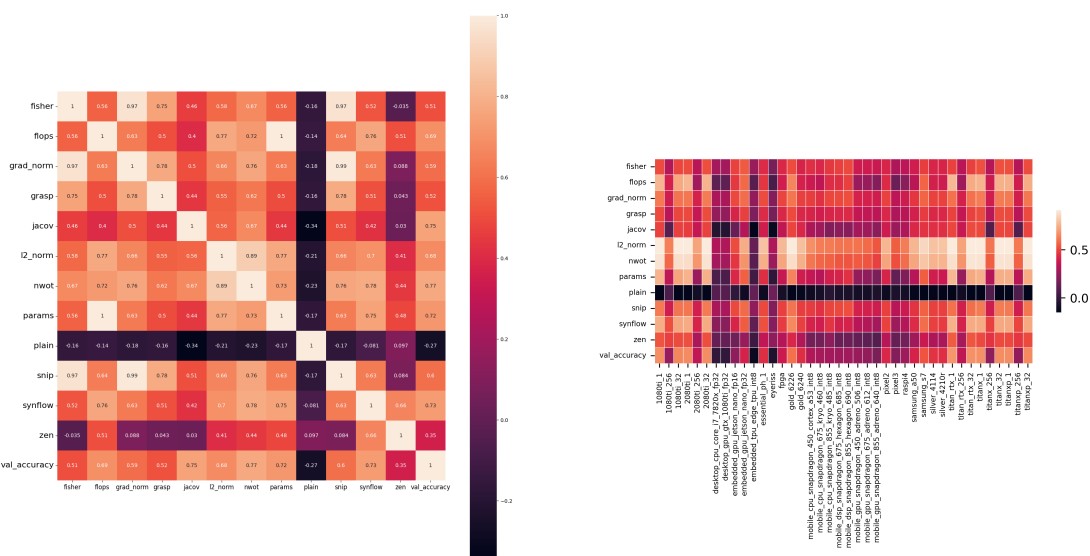

Figure 12: Correlation between zero cost proxies utilized in our ZCP encoding for the NASBench-201 CIFAR-10 search space.

Figure 13: Correlation between Zero Cost Proxies and device latencies.

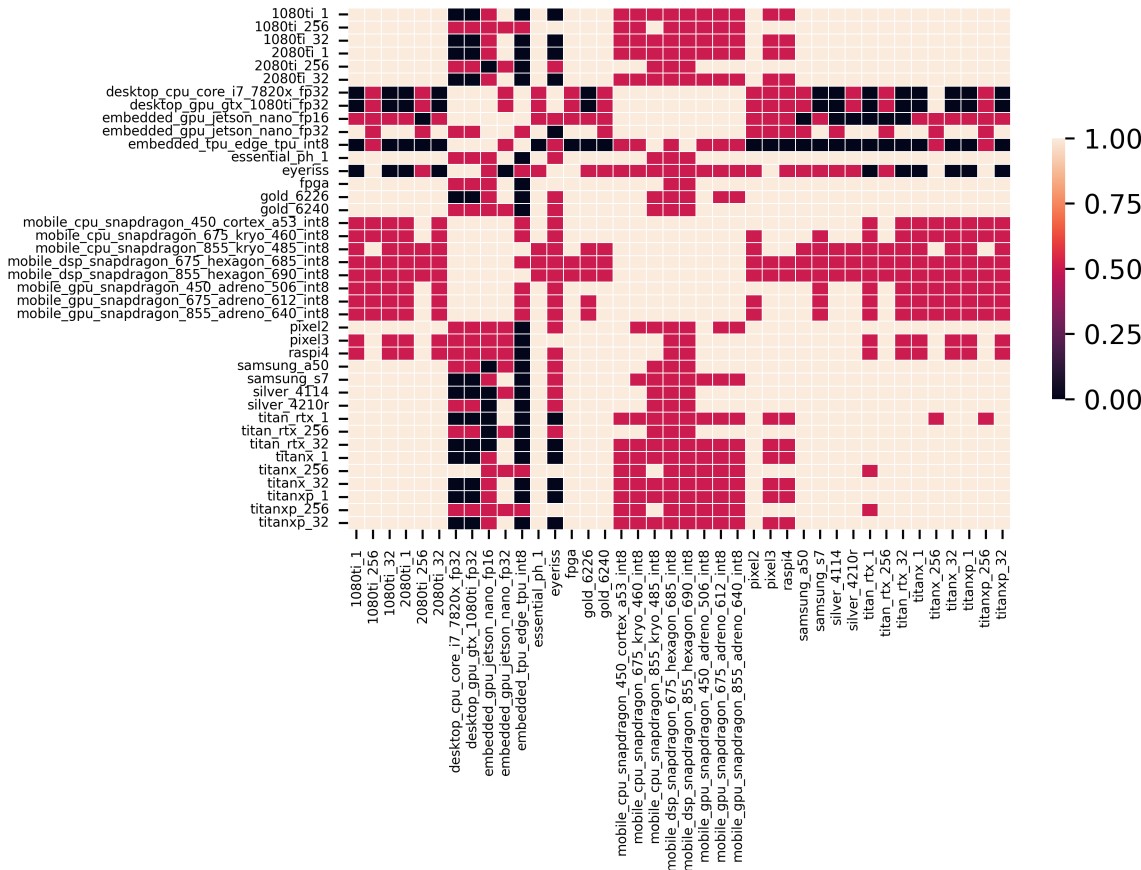

Figure 14: Correlation between hardware device latencies on the NASBench-201 search space. 1 represents the device correlation is greater than 0.7, 0.5 indicates that the device correlation is greater than 0.5, but lesser than 0.7, finally, 0 indicates that the device correlation is less than 0.5. We find that most devices have a corresponding highly correlated device.

| FBNet Default Train-Test Device Correlations | | | |
|---|---|---|---|
| | Test Device | | |
| Train Device | eyeriss | fpga | raspi4 |
| 1080ti_1 | 0.25 | 0.23 | 0.26 |
| 1080ti_32 | 0.53 | 0.42 | 0.52 |
| silver_4114 | 0.62 | 0.67 | 0.64 |
| silver_4210r | 0.63 | 0.68 | 0.65 |
| essential_ph_1 | 0.68 | 0.7 | 0.66 |
| samsung_s7 | 0.69 | 0.71 | 0.68 |
| 1080ti_64 | 0.76 | 0.61 | 0.72 |
| samsung_a50 | 0.86 | 0.86 | 0.84 |
| pixel3 | 0.98 | 0.91 | 0.96 |
| $\rho$ Of Closest Train Device | **0.98** | **0.91** | **0.96** |
| HELP Predictor $\rho$ | 0.94 | 0.89 | 0.90 |

Table 9: The default task training device set of FBNet has devices whose correlation is higher than predictor $\rho$.

| NB201 Default Train-Test Device Correlations | | | |
|---|---|---|---|
| | Test Device | | |
| Train Device | eyeriss | fpga | raspi4 |
| 1080ti_1 | 0.42 | 0.83 | 0.65 |
| 1080ti_32 | 0.43 | 0.84 | 0.67 |
| samsung_s7 | 0.52 | 0.89 | 0.76 |
| essential_ph_1 | 0.62 | 0.92 | 0.81 |
| silver_4114 | 0.59 | 0.94 | 0.84 |
| samsung_a50 | 0.63 | 0.96 | 0.87 |
| silver_4210r | 0.62 | 0.97 | 0.88 |
| 1080ti_256 | 0.89 | 0.89 | 0.73 |
| pixel3 | 0.72 | 0.87 | 0.97 |
| $\rho$ Of Closest Train Device | 0.89 | 0.97 | **0.97** |
| HELP Predictor $\rho$ | **0.94** | **0.99** | 0.89 |

Table 10: The default task training device set of NASBench-201 has devices whose correlation is very close to predictor $\rho$.

Table 11: The default device set has a high training-test device correlation (low task distance).

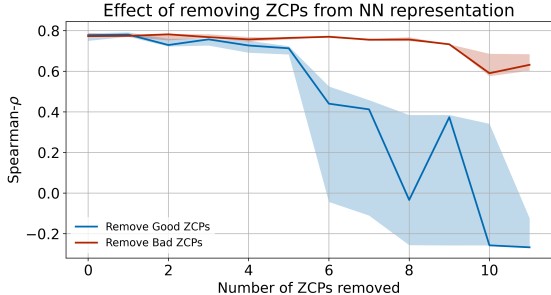

Figure 15: We train accuracy predictors with ZCP NN representation. We remove certain ZCPs from the NN representation and observe its effect on Spearman-$\rho$ of the accuracy predictor. We find that removing good ZCPs can have a worse impact over removing bad ZCPs for accuracy prediction on NASBench-201.

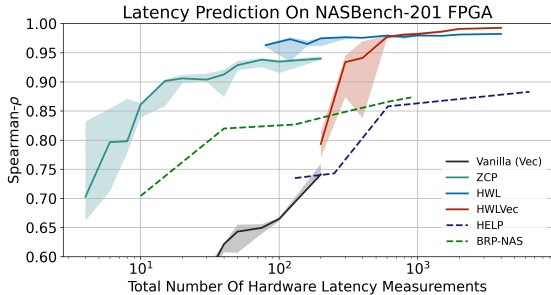

Figure 16: Training from scratch with ZCP/HWL encodings and comparing it with BRP-NAS and HELP

## A.8 Few Shot Latency and Accuracy Prediction

In Figure 17, we present the entire graphs for different NN encodings. We find that ZCP and HWL are generally the most sample efficient. Eyeriss has a low ZCP-latency correlation, thus Vec performs better in this case. However, we still see significant benefit from using ZCPVec, which indicates that modifying the NN encoding in exisiting tasks to include zero cost proxy evaluation can help us train better predictors at almost no extra cost.

## A.9 Multi-Search-Space-Task Accuracy Predictor Transfer

We train individual predictors from scratch on every task from TransNASBench-101 and NASBench search space using 15% of the architectures on the individual train space, and transfer it to every other task on the TransNASBench-101 and NASBench search space with only 10 samples. This transfer task utilizes the ZCP NN encoding, and is compared with a train from scratch strategy (on the new task only) with the Vec NN encoding. From Figure 19 (Left), we see a benefit of multi-search-space and multi-task transfer learning, the exact improvement is depicted in Figure 19 (Right).

## A.10 Limitations

**Representation Correlation**. We find that the domain agnostic NN encoding works extremely well for building few-shot predictors for latency and accuracy. However, our test in Figure 15 indicates that the effectiveness of this encoding is somewhat dependent on the correlation of the entries of ZCP and HWL with the predictor output. Fortunately, there exist several effective ZCPs that can serve as proxies for few-shot learning. Further, this highlights the importance of building a robust hardware benchmark such that new hardware can be adapted from predictors of previous benchmarks, as such encodings can be transferred across tasks and search-spaces.

|  | FPGA | RasPi4 | Eyeriss | Average |
|---|---|---|---|---|
| Default Set | 0.886 | 0.895 | 0.939 | 0.91 |
| Spearman < 0.7 | 0.663 | 0.689 | 0.727 | 0.69 |
| Spearman < 0.6 | 0.328 | 0.395 | 0.437 | 0.39 |
| Spearman < 0.3 | 0.146 | 0.159 | 0.201 | 0.17 |

Table 12: In this test, we remove devices with high correlation such that remaining devices have correlation lesser than the specific number. We find that removing highly correlated devices causes predictor performance to decrease extremely fast on the FBNet search space.

| | NB201-Adversarial Task. | | FBNet-Adversarial Task |
|---|---|---|---|
| Train | titan_rtx_1,titan_rtx_32,titanxp_1,2080ti_1,titanx_1,1080ti_1, titanx_32,titanxp_32,2080ti_32,1080ti_32,gold_6226,samsung_s7, silver_4114,gold_6240,silver_4210r,samsung_a50,pixel2 | Train | 1080ti_1,1080ti_32,1080ti_64,2080ti_1,2080ti_32,2080ti_64,, titan_rtx_1,titan_rtx_32,titan_rtx_64,titanx_1, titanx_32,titanx_64,titanxp_1,titanxp_32,titanxp_64 |
| Test | eyeriss,desktop_gpu_gtx_1080ti_fp32,embedded_tpu_edge_tpu_int8 | Test | gold_6226,essential_ph_1,samsung_s7,pixel2 |

Table 13: Adversarial device sets for the NASBench and FBNet search spaces. As seen here, most of the training devices for FBNet are GPU, as they exhibit low correlation with the test devices. Our NB201-Adversarial Task exhibits more diversity, as we combine the EAGLE [5] and HELP [8] HW latency data-sets.

**HW Embedding Correlation**. While we introduce a sample-free method of representing hardware devices, the effectiveness of these embeddings largely rely on the diversity of the HW-NN samples that the predictor is trained on. Further, the additional sample efficiency offered by our embedding initializer in Eq. 1 depends on the diversity of the hardware platforms the predictor was trained on.

| Transfer Learning Architecture and Hyper Parameters | |
|---|---|
| Cross Domain Network Layer Size | 128 |
| Cross Domain Network Depth | 4 |
| Optimizer | AdamW |
| Pre-Training Optimizer LR | 0.004 |
| Scheduler | CosineAnnealingLR |
| Training Epochs | 250 |
| Transfer Epochs | 50 |
| Transfer Optimizer LR | 0.0004 |
| Optimizer Weight Decay | 0.0005 |

Table 14: Hyper-parameter configuration for Accuracy Transfer Learning results.

| Transfer Learning Architecture and Hyper Parameters | |
|---|---|
| Source Hardware Samples | 900 (NB201)/4000 (FBNet) |
| Batch Size | 128 |
| Source Hardware Epochs | 250 |
| Target Hardware Epochs | 50 |
| HW Embedding Size | 8 |
| GCN Layer Size (NB201) | 200 |
| GCN Layer Depth (NB201) | 3 |
| GCN FC Layer Size | 200 |
| GCN FC Layer Depth | 3 |
| NN Layer Size (FBNet) | 100 |
| NN Layer Depth | 3 |
| NN Embedding Dimension | 30 |
| Interaction Feature Size | 1000 |
| Interaction Feature Depth | 2 |
| Optimizer | AdamW |
| Source Optimizer LR | 0.0004 |
| Target Optimizer LR | 0.001 |
| Optimizer WD | 0.0005 |

Table 15: Hyper-parameter configuration for Hardware Transfer Learning results.

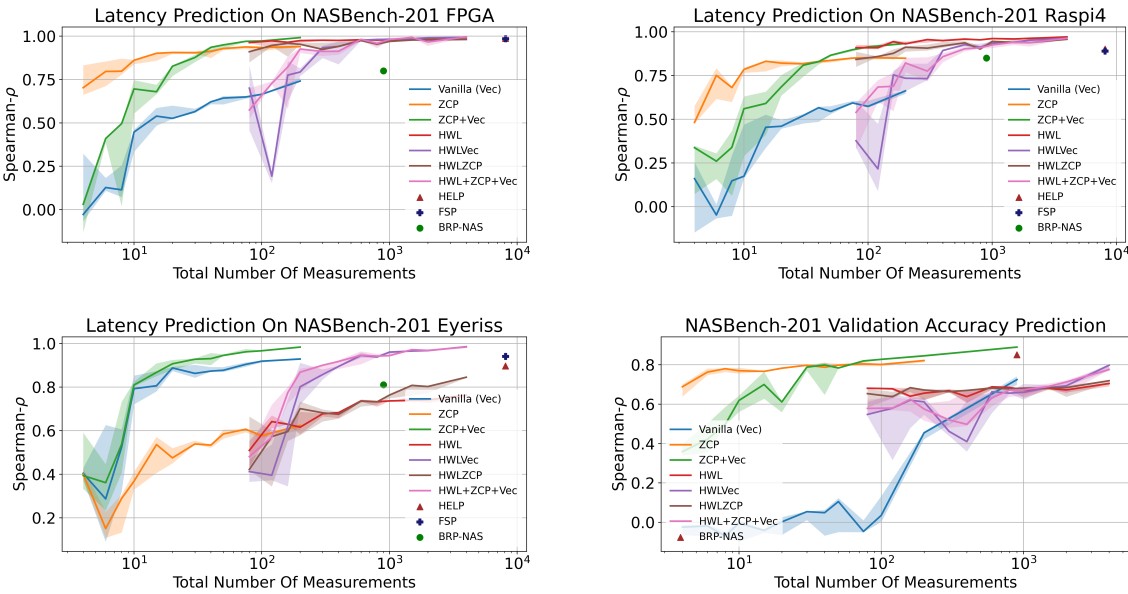

Figure 17: Total number of HW Measurements and Trained Model Samples required to predict latency and accuracy on NASBench-201 for various NN encodings, with respect to HELP and Trasnfer Learning (FSP).

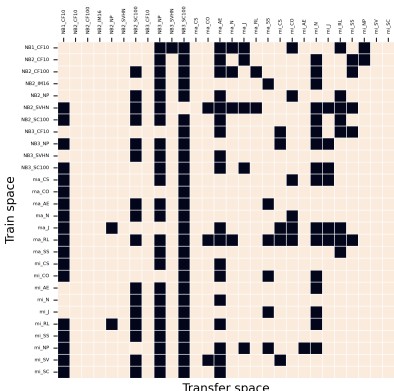

Figure 18: The white boxes indicate improvement in performance when using ZCP with transfer learning.

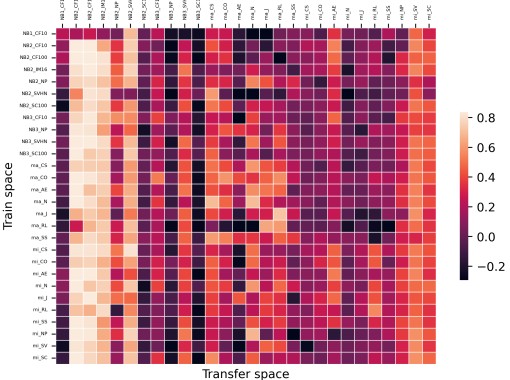

Figure 19: Presents the heatmap for the improvement in performance when using ZCP with transfer learning over Vec train-from scratch.

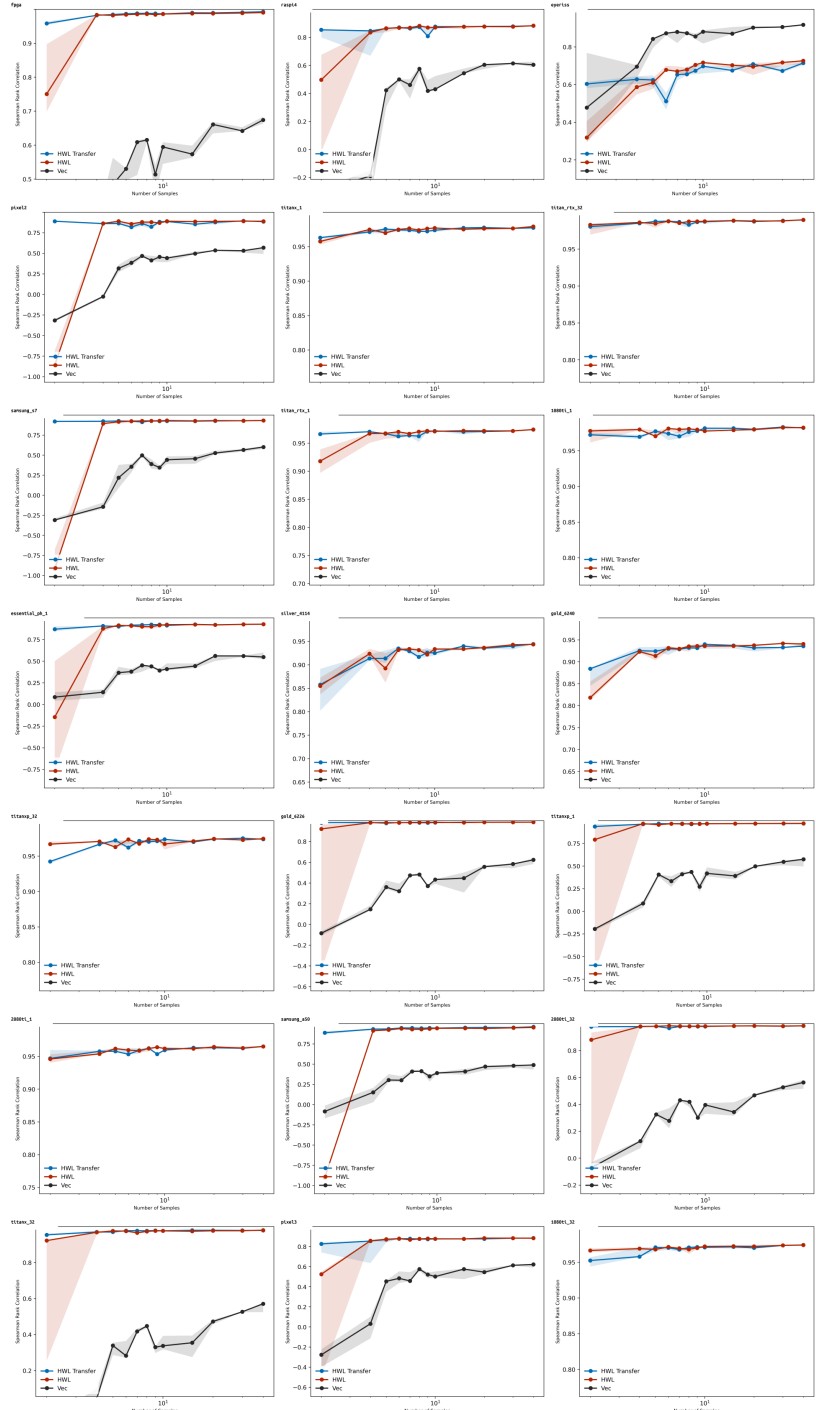

Figure 20: For FBNet to NASBench-201, HWL Transfer improves sample efficiency, indicating that we can leverage transfer learning for multi-search-space latency prediction. (Vec line not visible in some graphs as its performance is too poor, and we set the Y scale to focus on HWL and HWL Transfer.

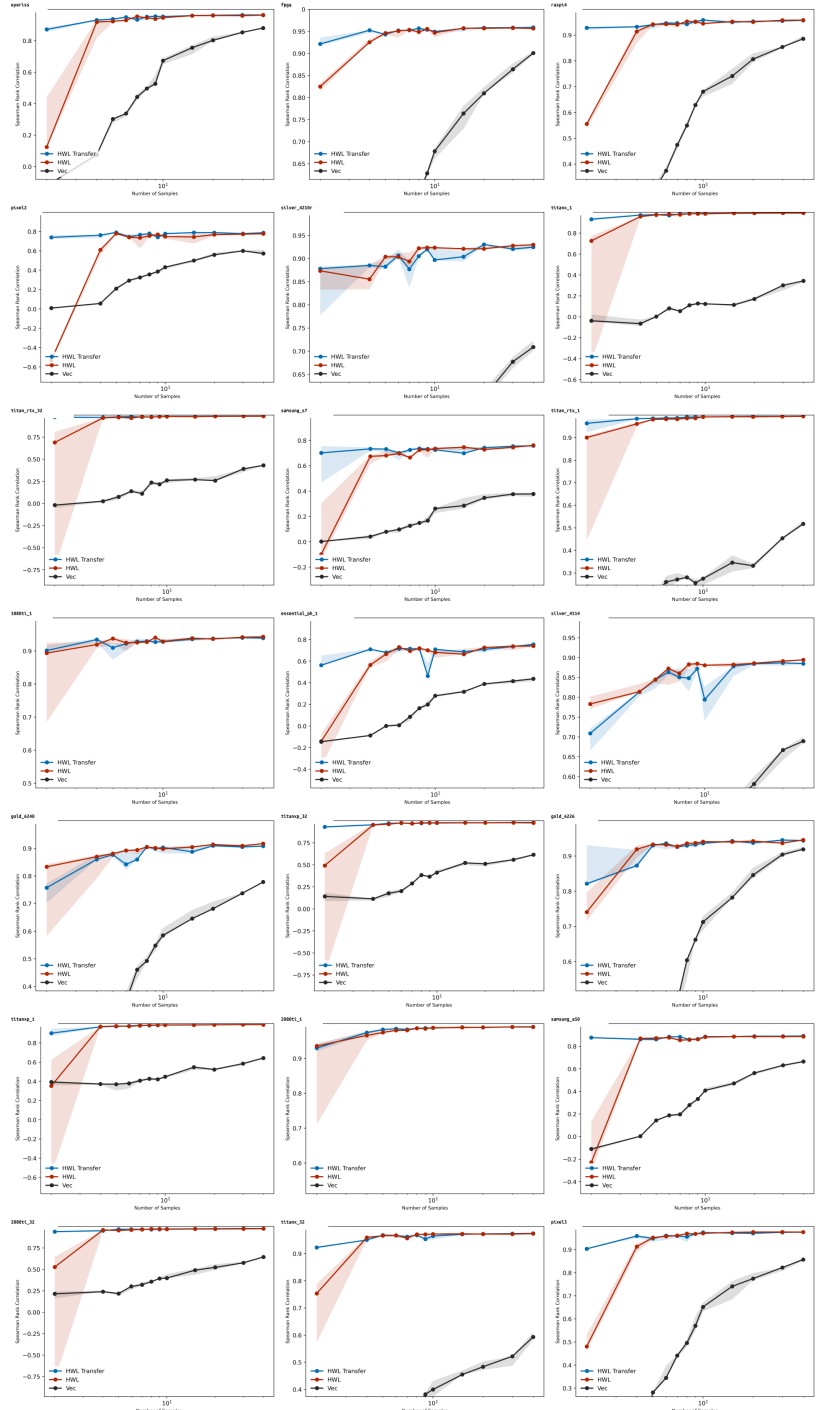

Figure 21: For NASBench-201 to FBNet, HWL Transfer improves sample efficiency, indicating that we can leverage transfer learning for multi-search-space latency prediction. (Vec line not visible in some graphs as its performance is too poor, and we set the Y scale to focus on HWL and HWL Transfer.

