# OpenReview forum: "Multi-Predict: Few Shot Predictors For Efficient Neural Architecture Search"
_automl.cc/AutoML/2023/Conference — AutoML 2023 MainTrack_

### Official Review · Reviewer_NUxr · 2023-04-12

**Potential Impact On The Field Of Automl Rating:** 3
**Technical Quality And Correctness Rating:** 2
**Clarity Rating:** 3
**Actions Required To Increase Overall Recommendation:** Please consider addressing my concern…

**Summary Of Contributions:**

The paper provides a few-shot multi-search-space latency and accuracy predictors that could be used to expedite NAS methods. The paper focuses on generalizing the task of prediction such that it is agnostic to task, search space, and hardware. I feel this is the most interesting contribution from the paper - Instead of using a vector for representing NNs, authors leveraged a search-space independent network embedding vector that is based on either zero-cost proxies or hardware latency measurements.

**Clarity:**

 The paper is well-written and easy to follow. I have some minor concerns that addressing them could improve the paper's presentation.

o    Line 47: Typo! “devices.While…”-> “devices. While…”

o    Line 72: “we’d…”-> “we would…”

o    Related work should be extended to cover recently published related papers proposing latency predictors for CNNs, e.g., Loni, M., Zoljodi, A., Majd, A., Ahn, B. H., Daneshtalab, M., Sjödin, M., & Esmaeilzadeh, H. (2021). Faststereonet: A fast neural architecture search for improving the inference of disparity estimation on resource-limited platforms. IEEE Transactions on Systems, Man, and Cybernetics: Systems, 52(8), 5222-5234.

o    Line 99: “Most prior work [???] has focused on using ZCPs as a weak predictor of accuracy within a NAS search.”-> Citation(s) is needed.

o    I am not sure if this paper needs a separate section for the ablation study. I expect the authors to clarify if they represent the impact of each individual contribution.

o    Figure 6: It is not clear in the figure legend which of these lines is proposed by the authors. Plus, I recommend authors cite each method in the figure legend.

**Overall Review:**

•	Strengths:

o	It is a relevant topic for the conference.

o	The message of the paper is well supported.

o	This paper addresses a SoTA problem in the AutoML domain.


•	Weaknesses:

o	Computational cost and carbon footprint of the accuracy and latency predictors have not been reported.

o	The manuscript contains a few typos.

o	A multi-device latency predictor may not work in practice.


**Potential Impact On The Field Of Automl:**

The majority of AutoML routines suffer from expensive training loops. Thus, model performance (e.g., latency) and accuracy estimation are becoming prevalent in AutoML, which substantially reduces benchmarking costs but increases the risk of misleading results. A major benefit of this work is to improve the reliability of estimating model performance and accuracy.

**Reproducibility (Optional):**

There is a code available for the paper.

**Review Confidence:**

5: You are absolutely certain about your assessment. You are very familiar with the related work and checked all the details carefully.

**Review Rating:**

8: Accept: Technically sound paper with major impact and strong evaluation, with perhaps some minor flaws.

**Review Summary:**

Overall the idea of using a novel network embedding for building design-space and task-agnostic accuracy/latency predictor is quite interesting. I give this paper a weak accept, but I'm willing to change my evaluation if the mentioned concerns are addressed.

**Technical Quality And Correctness:**

·         Overall, the approach and experimentation are technically correct and sound. However, I have serious concerns about the new research trend of proposing multi-device latency predictors. I believe it is a huge claim to say we have a latency predictor that works for any device, even with some learning samples. It is not only about the method proposed in this paper but also other related papers such as HELP [1]. To be honest, I feel the ML/AI community neglects details of the hardware backend and the compilation pipeline. The main reason is that HW-NAS-Bench results are obtained for a certain HW configuration, e.g., the cuDNN version for GPUs is 10.0. Predicting the latency of FPGAs is more difficult, as many different parameters are involved in synthesizing a NN on FPGA, such as quantization level, synthesizer optimization objective (area, latency), routing objectives, and many others. In addition to synthesizer parameters, if we change the FPGA acceleration method, for example using systolic arrays [2] or a high-level synthesizer [3], I am pretty sure the latency predictor does not work. For a simple CPU, you can compile Numpy using the Intel MKL library [4] or non-SIMD instructions (default). Therefore, I believe we need to consider the variations of all these configurations for each device in our dataset.

o    [1] LEE, H., Lee, S., Chong, S., and Hwang, S. J. (2021). Help: Hardware-adaptive efficient latency prediction for nas via meta-learning. In 35th Conference on Neural Information Processing Systems (NeurIPS) 2021. Conference on Neural Information Processing Systems (NeurIPS).

o    [2] Wei, Xuechao, et al. "Automated systolic array architecture synthesis for high throughput CNN inference on FPGAs." Proceedings of the 54th Annual Design Automation Conference 2017. 2017.

o    [3] Noronha, Daniel H., Bahar Salehpour, and Steven JE Wilton. "LeFlow: Enabling flexible FPGA high-level synthesis of tensorflow deep neural networks." FSP Workshop 2018; Fifth International Workshop on FPGAs for Software Programmers. VDE, 2018.

o    [4] https://www.intel.com/content/www/us/en/developer/articles/technical/numpyscipy-with-intel-mkl.html

I am looking forward to hearing the authors’ opinions regarding this concern.

·         ZiCO [5] is a zero-shot accuracy proxy. Please consider discussing the differences between this paper and ZiCO.

o    [5] Li, Guihong, et al. "ZiCo: Zero-shot NAS via Inverse Coefficient of Variation on Gradients." arXiv preprint arXiv:2301.11300 (2023).

·         Table 1: ‘TL ZCPVec ET’ provides better results than ‘TL HWLVec ET’. Why?

·         Line 113: What is the advantage of selecting a four-layer MLP as the regression model? Did the authors use an automated method to select the most suitable regression model?

·         The computational cost and carbon footprint of the accuracy and latency predictors have not been reported.

---

> ### Author Response · Authors · 2023-05-02
> **Response to Reviewer NUxr**
>
> We would like to thank the reviewer for their insightful comments.
>
> > ‘... I believe it is a huge claim to say we ... even with some learning samples. A multi-device latency predictor may not work in practice….’
>
> We acknowledge that there are several HW designs that would be sufficiently un-correlated with the training set of devices such that pre-training does not work as effectively. In fact, our construction of the ‘Adversarial device set’ is an explicit attempt to model situations where the ‘test’ device is novel and difficult to model using existing reference devices. We do not intend to claim that we have a latency predictor that works for any device, rather claim that we have improved the sample efficiency of existing latency predictors, specifically by using HWL and ZCP representations and learnable HW embeddings. A multi-device latency predictor may not work in practice, but our tests indicate that transfer of knowledge from existing search spaces, even if the correlation is not very high, still benefits end to end sample efficiency of the prediction task. Kindly note that we do not have a ‘multi-device’ latency predictor, rather, we have a baseline predictor that works on a set of training devices, and is adapted to the specific test device. This adaptation process results in a predictor that is not generalized, and works only on the specific target HW . Additionally, we consider a device to be a combination of HW and software, so if the software changes, it would be a different device in our characterization. We completely agree that predicting the latency on FPGAs with different accelerator parametrization, quantization levels and routing objectives are more difficult. In our case, by FPGA, we refer to a specific accelerator on the fpga, not the unconstrained search space of possible accelerators. again, each of those would be considered a separate device in our methodology. We follow the convention of prior work HELP and refer to a single instantiation of the accelerator as an FPGA to keep it easy to compare results.
> We agree that multi-device prediction is not a trivial task, and an extremely novel HW device may be distinct enough to make transfer learning less effective. Nevertheless, we take steps towards increasing the sample efficiency of existing methods for few-shot latency and accuracy prediction. We believe that as an increasing catalogue of HW latencies is made available to us, in conjunction with our search space agnostic representation of NN architectures, it would be possible to build unified datasets with a very diverse device set, making few-shot latency prediction on HW devices more feasible in the future.
>
> > ‘... Please consider discussing the differences between this paper and ZiCO. …’
>
> We believe that the performance of accuracy predictors would improve if more ZCPs are added. We adapt prior work NAS-Bench-Suite-Zero to obtain the ZCPs for our studies. We will discuss ZiCO in the related work.
>
> > ‘...  ‘TL ZCPVec ET’ provides better results than ‘TL HWLVec ET’. Why? …’
>
> We find that some of the ZCPs have a higher correlation than the reference training devices with the test devices. This is in agreement with our analysis in Figure 15 (Appendix) where we discuss the effectiveness of input-output correlation for predictors.
>
> > ‘... most suitable regression model? …’
>
> We adapt predictors from BRP-NAS and HELP for consistency with prior work. A separate study on predictors (XGB, GP etc.) is warranted, but we focus on representations in this paper. We believe that a study on the interaction between predictors, representations and search algorithms warrants a separate, in depth investigation. However, we have added NAS search results to demonstrate the end-to-end effectiveness of our representation. Please see response to reviewer H8rP for more on this.
>
> > ‘...  The computational cost and carbon footprint of the accuracy and latency predictors have not been reported. …’
>
> It takes approximately 10 GPU minutes to train a predictor for accuracy and latency. These predictors are extremely sample efficient to train. Our paper focuses on training efficient predictors for NAS, and we consider the carbon footprint of the training process to be negligible when compared to other methods.
>
> > ‘... this paper needs a separate section for the ablation study. I expect the authors to clarify if they represent the impact of each individual contribution.  …’
>
> We investigate several aspects of few-shot regression of latency and accuracy in this paper. To introduce the terminology and methodology in a complete fashion, we conduct some ablation studies and include them in the paper. We further support our claims with more experiments in the Appendix.
>
> > ‘... figure legend which of these lines is proposed by the authors. Plus, I recommend authors cite each method in the figure legend….’
>
> We have cited the respective methods in Figure 6, and will regenerate all figures to include appropriate citations.

---

> > ### Comment · Reviewer_NUxr · 2023-05-02
> > **Response**
> >
> > Thanks for addressing my concerns. I increased my score from 7 to 8.

---

### Official Review · Reviewer_Updo · 2023-04-12

**Potential Impact On The Field Of Automl Rating:** 3
**Technical Quality And Correctness Rating:** 2
**Clarity Rating:** 3

**Summary Of Contributions:**

This paper tackles the problem of search space-dependent performance prediction models used so far in NAS. This paper proposes a search space-independent neural network encoding based on zero-cost proxies and hardware embeddings. This way, the proposed method shows an improved correlation for latency prediction in comparison to SOTA baselines. Furthermore, this method enables transfer to other unseen devices and search spaces as well as tasks for latency and accuracy prediction. The paper evaluates the proposed method on NAS-Bench-201 and FB-NET for different devices.

**Actions Required To Increase Overall Recommendation:**

In order to increase the overall recommendation, this paper would need to answer the open questions in Technical Quality And Correctness.

**After Rebuttal:** I increase my score from 4 to 6.

**Clarity:**

Line 53-54: “improve […] by more than an order of magnitude”: very vague statement. The corresponding Figure 1 only shows improved correlation but no accuracy.

Line 72: “as few as 4 samples“: What and where is the according experiment?

Task distance mentioned in line 121 is not clear. Line 218 phrases is as a correlation between training and test devices, but it is not clear to me between what training and test device information this correlation is calculated.

Why are the hardware embedding needed in general? Can we not just use the NN encoding with ET? The focus of this paper is more on the hardware embedding than on the actual contribution of multi-search approaches.

Section 3.1 (1) sample: It is not clear to me what that means.

Section 3.2 HWL (lines 165-168): A connection to the hardware embedding as presented in Section 3.1 would be helpful here.

Line 178: “values of this row are assigned to be the same as a device h^{\tau}^{1}”: It is not clear what that means.

Line 189: What are x^{\tau}^1 and  x^{\tau}^2?


Section 4  few-shot latency predictor (lines 234-253):
This section does not clearly describe on what data the predictors are pretrained and for how long and how large this dataset is, and also not what the test data set is.

Line 241: “less”: The actual deltas are not that different given their correspodning starting points.

Figure 9: The diagonal should only contain 0 improvement, which is not the case. Also in what terms is the improvement ccalculated Correlation or accuracy? That is not clear here.


This paper also misses recent predictor based references:
1.	Junru Wu, Xiyang Dai, Dongdong Chen, Yinpeng Chen, Mengchen Liu, Ye Yu, Zhangyang Wang, Zicheng Liu, Mei Chen, Lu Yuan: Stronger NAS with Weaker Predictors. NeurIPS 2021
2.	Jovita Lukasik, Steffen Jung, Margret Keuper: Learning Where to Look - Generative NAS is Surprisingly Efficient. ECCV 2022

Small typos:
Line 96: Turner et al.
Line 98: Krishnakumar et al.


**Overall Review:**

This paper introduces an important research aspect.
It is mainly well-written and motivates the proposed approach in a good way.

However, there are some open question, which would need to be answered.
Especially, my main concern is, that there is no actual search result reported in this paper, but only correlations, which does not show the ability to find high-performing architectures.


**Potential Impact On The Field Of Automl:**

This work tackles an important research aspect, that search space-dependent predictors are used so far.

**Review Confidence:**

5: You are absolutely certain about your assessment. You are very familiar with the related work and checked all the details carefully.

**Review Rating:**

6: Borderline Leaning Accept: Technically sound paper where reasons to accept outweigh reasons to reject. Please use sparingly.

**Review Summary:**

The idea of taking a step forward and using search space-independent predictors is great, I believe more results on actual search results and more common NAS-Benchmarks are needed.



**Technical Quality And Correctness:**

In order to assess the technical quality of this paper, I have some open questions:

Section 5: No accuracy values are shown and associated with this no search results in terms of found architectures and their accuracies.

Where is the latency information from?
The paper states in lines 21 and 98, that experiments on 28 NAS search spaces and tasks were conducted? The main paper shows experiments on only 3 different search spaces (FBNet, NAS-Bench-201 and TransNASBench-101), which does not sum to 28?

In addition, this paper does not compare to other known benchmarks, which are also provided in Meta et al.

Line 75: Which 95 domains?

How many runs were conducted for all tables?

Yash Mehta, Colin White, Arber Zela, Arjun Krishnakumar, Guri Zabergja, Shakiba Moradian, Mahmoud Safari, Kaicheng Yu, Frank Hutter: NAS-Bench-Suite: NAS Evaluation is (Now) Surprisingly Easy. ICLR 2022.

---

> ### Author Response · Authors · 2023-05-02
> **Response to Reviewer Updo**
>
> We appreciate the reviewer's feedback and acknowledge the concerns raised. We would like to address these points and provide clarification to enhance the understanding of our work.
>
> > ‘... Section 5: No accuracy values are shown and associated with this no search results in terms of found architectures and their accuracies….’
>
> In this paper, we study two ways of representing neural networks, transfer learning of predictors, and hardware embeddings. We take a first step in evaluating the effectiveness of these methods from a prediction-based NAS perspective. We agree that including test data-set accuracies on NAS tasks could be beneficial for future comparisons. To this end, we add a ‘NAS Search Effectiveness’ section [Line 315] to our accuracy evaluation to clearly compare between Vec and ZCP.  We choose one of the state of the art iterative selection strategy introduced in BRP-NAS to evaluate our search effectiveness. (Refer to response to reviewer H8rP). We believe that there is an interaction between search algorithms and the input representations, which warrants a separate study.
>
> > ‘... Where is the latency information from?  …’
>
> We detail our latency sources in Appendix A.2.
>
> > ‘...which does not sum to 28?...’
>
> We say we test transfer of predictors on 28 NAS search spaces AND tasks. TransNASBench101 Micro and Macro have different sets of tasks, and NB201 and NB301 have different datasets, as detailed in Table 8. Our accuracy transfer matrix in the Appendix (Figure 17) studies transfer across all of these search spaces and tasks. There is limited correlation between tasks in the TB101 space, which makes the transfer task challenging.
>
> > ‘... , which are also provided in Meta et al….’
>
> The paper provides results on 'NAS Methods' focusing on the ranking ability of NAS Algorithms, whereas we look at representations for MLP prediction-based NAS. We believe a study on the effectiveness of different NAS Algorithms (RS/RE, etc.) and Predictors (GP, XGB, etc.) warrants a separate manuscript. For transfer, the Meta et al. paper looks at the generalizability of hyperparameters, and not the models themselves.
>
> > ‘... Line 75: Which 95 domains? …’
>
> Figure 9 Micro -> Macro transfers (10*7) = 70
> NB2CF10, NB2CF100, NB2IM16, NB3CF10, NB3SVHN transfers (5*5) = 25
> 70 + 25 = 95
>
> > ‘...How many runs were conducted for all tables?...’
>
> 5 runs were conducted for all the tables. We will include this information in the appendix. Thank you.
>
> > ‘... more than an order of magnitude”: very vague statement. The corresponding Figure 1 only shows improved correlation but…’
>
> We refer to improving the sample efficiency of the predictors without compromising the effectiveness of the predictors. We improve correlation with fewer samples.

---

> > ### Author Response · Authors · 2023-05-02
> > **Continued Response to Reviewer Updo**
> >
> >
> > > ‘... Line 72: “as few as 4 samples“: What and where is the according experiment? …’
> >
> > This information can be found in Figure 7 (Multi-Search-Space Latency Prediction). It demonstrates how effective a good representation can be for both few-shot learning and few-shot transfer of hardware predictors.
> >
> > > ‘...  it is not clear to me between what training and test device information this correlation is calculate …’
> >
> > Thank you for the note, we have clarified it on Line 225. The latency/accuracy correlation between two devices is the task distance.
> >
> > > ‘... The focus of this paper is more on the hardware embedding than on the actual contribution of multi-search approaches….’
> >
> > Hardware embedding enables higher sample efficiency for multi-hardware latency prediction tasks. The hardware embedding (which actually is represented as an embedding table) enables a good hardware device description initialization, which improves sample efficiency.
> >
> > > ‘... (1) sample: It is not clear to me what that means….’
> >
> > Thanks for pointing it out, we have attempted to clarify this explicitly in Section 3.1 [Line 140]. To represent a device, an 'n' dimensional vector is used. This 'n' dimensional vector is populated with latency measurements of 'n' fixed neural networks on the device.
> >
> > > ‘... h^{\tau}^{1}”: It is not clear what that means. …’
> >
> > We search for the device that has the highest correlation with the 'new' device, and use its embedding as initialization for the new device.
> >
> > > ‘... What are x^{\tau}^1 and x^{\tau}^2? …’
> >
> > The dimensions of the vector representation of the neural network may not be the same. x^{\tau}^1 is the 'vec' representation of a sample NN architecture in \tau^{1} search space. (Line 157: Vec Encoding (X^{\tau})
> >
> > > ‘...does not clearly describe on what data the predictors are pretrained and for how long …’
> >
> > The predictors are pre-trained on 900/4000 latency samples on a training device set. The size of the NASBench-201 dataset is 15625, and size of the FBNet dataset is 5000. The test data is all the remaining architectures which are not used for training. We have clarified this in Section 4 [Line 219].
> >
> > > ‘...  not that different given their corresponding starting points. …’
> >
> > We agree, in general, we remark that pre-training is important.
> >
> > > ‘... Also in what terms is the improvement calculated Correlation or accuracy?  …’
> >
> > The diagonal has a non-zero improvement because ZCP predictors are inherently better without transfer learning. Improvement is the difference in spearman rank correlation. We have made this clarification in Figure 9, thank you for pointing it out.
> >
> > > ‘... which does not show the ability to find high-performing architectures. …’
> >
> > We focus on the representation in this paper, test data-set accuracies would require us to do a robust study on a large set of search algorithms, which we believe is a separate study to assess the interaction between input representation and search strategy.
> >
> > We have made major revisions to this paper to address the concerns of the reviewer. We believe that our search space agnostic representation can significantly improve the sample efficiency of NAS for both accuracy and latency, and would request the reviewer to kindly reconsider their scores in light of our revision and clarifications.

---

> > > ### Comment · Reviewer_Updo · 2023-05-08
> > > **Response to Authors**
> > >
> > > I thank the authors for the responses. However, I disagree that showing the test accuracies of the found architectures from NAS benchmarks (therefore only querying the architectures) is a separate study. In order to access the ability to find well-performing architectures in terms of hardware and accuracy both information is needed. I appreciate, that the authors included a section *NAS Search Effectiveness*. This section leads to more questions, about how a test accuracy of 100% in NAS-Bench-101 is possible when the optimum is 94.32% (the same holds for NAS-Bench-201).
> > > I will slightly raise my score. Though some of my concerns and questions were answered, I still believe that this paper needs more information about the found architectures in terms of accuracy.

---

> > > > ### Author Response · Authors · 2023-05-08
> > > > **Response To Reviewer**
> > > >
> > > > Dear Reviewer,
> > > >
> > > > Thank you for your valuable feedback.
> > > >
> > > > We would like to clarify that in the caption of Figure 10, we had specifically stated that the Accuracy is normalized within the range of 0 to 1. To eliminate any confusion, we will revise the figure to display the un-normalized test accuracy rather than the normalized version. As the rebuttal period has expired, we are unable to update the paper with the new figure. Please find this anonymized link to the figure here: https://ibb.co/MVrSCSy
> > > >
> > > > It is worth mentioning that we employed a recent, highly-cited methodology for assessing the search performance of predictors (BRP-NAS) based on the ZCP representation. We evaluated the ZCP representation across approximately 423,000 neural networks within the NASBench-101 search space, thus establishing a fair benchmark for future comparisons. The updated figure now includes the precise accuracy of the discovered architectures. We agree that presenting the test accuracies of the identified architectures on NAS Benchmarks is essential, and have amended our manuscript accordingly.
> > > >
> > > > If there are any additional steps we can take to assist you in reevaluating your score, please let us know.

---

> > > > > ### Comment · Reviewer_Updo · 2023-05-08
> > > > > **Response to Reviewer**
> > > > >
> > > > > I appreciate the fast response of the authors and their response. I am willing to raise my score and encourage the authors to update the submission (in case of being a camera-ready version) by including the updated figure and the information about the test accuracies of the identified architectures.

---

### Official Review · Reviewer_H8rP · 2023-04-13

**Potential Impact On The Field Of Automl Rating:** 2
**Technical Quality And Correctness Rating:** 3
**Clarity Rating:** 3

**Summary Of Contributions:**

This paper proposes a novel NN encoding called Multi-Predict. The encoding is based on the combination between Zero Cost Proxies (ZCP), and Hardware Latencies (HWL) in order to improve Neural Architecture Search (NAS). The main differences between this paper and prior works is that this one uses a combined approach between ZCPs and HWLs to generate a search space agnostic method to predict neural architecture outcomes. The authors did a variety of comparisons against the BRP-NAS and HELP methods, using NASBench-201 as the primary comparison framework.

**Actions Required To Increase Overall Recommendation:**

I would recommend adding further experiments on the ZCPs in order to give a more holistic comparison against the baseline for this algorithm. Also, including accuracy values for some experiments will improve the paper’s ability for future comparisons.

**Clarity:**

The majority of this paper was clearly written. I had a fair bit of confusion reading through the first time trying to sort out the different methods being used, but overall it was acceptable.

**Overall Review:**

Strengths:

The authors introduce a new method for creating neural network encodings, and leveraging it to create better neural architectures. The motivation is clear, where we are looking to improve model latencies and performance simultaneously, while also keeping the usually large NAS costs down. To accomplish this, they encode information from a ZCP and a HWL in order to introduce a more efficient NAS search space.

Weaknesses:

Taking from the definition of this paper, there is a Zero Cost Proxy that should be used in this final method to perform NAS. While there is a basic definition of what a ZCP is, and how they are used, there is no explanation of these ZCPs outside of the appendix. There is also a lack of experimentation for the ZCPs. One such example would be to compare the base ZCP performance against your model. While we could assume that your model would perform better due to being an ensemble method, this is not explicitly mentioned in the paper.

I also feel that even though the Spearman's rank correlation is useful in describing the effectiveness of this method, showcasing test dataset accuracies on NAS tasks would be very useful for future comparison against this work.


**Potential Impact On The Field Of Automl:**

The impact of this work is strictly within the field of NAS, through the increased efficiency of using this new NN encoding. There is a distinct improvement over prior methods shown within this work for few-shot NAS in a hardware agnostic manner. However, this problem is unlikely to see major use within the field.

**Review Confidence:**

3: You are fairly confident in your assessment. It is possible that you did not understand some parts of the submission or that you are unfamiliar with some pieces of related work.

**Review Rating:**

4: Weak Reject: For instance, a paper with minor technical flaws, limited impact, and/or weak evaluation.

**Review Summary:**

Overall, this paper has concise experiments that prove the major points layed out in the abstract. The paper also clearly identifies its own contributions to the field, and how they are useful. However, this new NN encoding does not appear to have a high potential within the field of AutoML.

**Technical Quality And Correctness:**

This paper has clear and robust experimentation, and showcases a couple different comparisons on NASBench-201.

There are a few concerns with the overall breadth of the experimentation, but the depth of experimentation on NASBench-201 clearly showcased the algorithm's performance.

---

> ### Author Response · Authors · 2023-05-02
> **Response to Reviewer H8rP**
>
> We appreciate the reviewer's feedback and acknowledge the concerns raised. We would like to address these points and provide clarification to enhance the understanding of our work.
>
> Our work employs a set of Zero Cost Proxies (ZCPs), and while we understand that a detailed description of each ZCP would be helpful, including these descriptions within the main body of the paper is not feasible due to space constraints. However, we have provided references to the type and citation for each ZCP used in our study in Table 6 of the Supplemental material. This allows interested readers to explore the specific details of each ZCP in their respective sources.
>
> > ‘... compare the base ZCP performance against your model….’
>
> Thank you for pointing this out. Generally, our method improves over the best zero cost proxy significantly. Further, we have included all device - ZCP correlations in Figure 12 to serve as a reference point.
>
> > ‘...  showcasing test dataset accuracies on NAS tasks would be very useful for future comparison against  …’
>
> The primary focus of our paper is on representation. We agree that including test data-set accuracies on NAS tasks could be beneficial for future comparisons. To this end, we add a ‘NAS Search Effectiveness’ section [Line 315] to our accuracy evaluation to clearly compare between Vec and ZCP. We would like to remark that conducting a robust study on a large set of search algorithms to obtain test accuracies would require a separate investigation to assess the interaction between input representation and search strategy. As such, it is beyond the scope of our current work. However, we choose one of the state of the art iterative selection strategy introduced in BRP-NAS to evaluate our search effectiveness. We believe that our novel neural network encoding method and the experiments presented in the paper make significant contributions to the field of AutoML.

---

> > ### Author Response · Authors · 2023-05-08
> > **Response to reviewer**
> >
> > Dear Reviewer,
> >
> > As we approach the end of the author-reviewer discussion period, we kindly request your attention to review our revised manuscript and the responses to your comments as soon as possible. This will enable us to address any further queries or concerns you may have before the discussion period concludes.
> >
> > We have made several improvements to our manuscript to incorporate the valuable feedback provided by the reviewers:
> > 1. Expanded our literature survey with relevant citations.
> > 2. Made several clarifications regarding the methodology used as well as explained Sample Hardware Embedding in greater depth.
> > 3. Employed a highly-cited methodology for neural architecture search (BRP-NAS) to evaluate the effectiveness of our ZCP based accuracy predictors (Figure 10), and included a NAS Search Effectiveness sub-section detailing our search methodology and presenting specific NAS results.
> >
> > If our rebuttal satisfactorily addresses your concerns, we kindly request that you consider revising your rating of our submission accordingly.

---

### Review · Reproducibility_Reviewer_T9oa · 2023-04-13

**Completeness Of Code And Dataset Supplement Rating:** 3
**Usability And Ease Of Reproducibility Rating:** 3

**Actions Required To Increase The Reproducibility And Overall Recommendation:**

Providing requirements (library and python versions) will make it possible to verify the code. On a side note, it would also benefit from providing direct link to section, tables and figures (executing s files produces which tables etc.).

**Completeness Of Code And Dataset Supplement:**

The provided code seems complete, however lacks the requirements file and therefore challenging to execute. Furthermore, it is difficult to navigate the code and the association to various tables and figures of the paper. Some parts of instructions provided are not clear.

**Overall Reproducibility Review:**

The provided code and dats seems complete and consistent with major results of the paper. However, I could not run the code to verify them. I believe this can be solved by providing the libraries and versions to use. The instructions to run the code can be more clear.

**Review Confidence:**

3: You are fairly confident in your assessment. It is possible that you did not understand some parts of the submission or that you are unfamiliar with some pieces of the code or data.

**Review Rating:**

7: Weak Accept, all critical aspects are reproducibile with minor effort, and the remainder are likely reproducible with minor additional effort.

**Review Summary:**

I believe the code provided is complete, however I wan not able to verify if it can be run. Therefore, I am not able to confidently assess the provided code. Some aspect of instruction in the code are not very clear.

**Summary Of Necessary Code And Dataset Supplement:**

Implementation of various NN embeddings: vector of zero-cost proxies (ZCP), and a vector of HW latency measurements (HWL)  and hardware embedding: sample, index, Embedding table. CIFAR10 dataset and TransNASBench-101, NASBench-201, and NASBench-301benchmarks are used. The experiments are performed on various search spaces and hardware latency constraints.

**Usability And Ease Of Reproducibility:**

I was not able to run the provided codes. Unfortunately the  requirements to run are missing. By providing and installing  the correct libraries, the code should be executable.

---

> ### Author Response · Authors · 2023-05-02
> **Response To Reproducibility Reviewer T9oa**
>
> We would like to thank the reviewer for conducting the reproducibility check.
>
> We have included a requirements txt file that can enable execution of our code. We have also added references to the tables which the sections in our README correspond to. We hope that this helps in reproducing our experiments.

---

### Official Review · Reviewer_ur9J · 2023-04-13

**Potential Impact On The Field Of Automl Rating:** 2
**Technical Quality And Correctness Rating:** 2
**Clarity Rating:** 2
**Actions Required To Increase Overall Recommendation:** Make the evaluations more consistent …

**Summary Of Contributions:**

The paper proposes a method that encodes zero-cost proxies or hardware latencies to make sample-efficient predictions about the considered task, specifically accuracy or latency, of an architecture. Majority of the paper focuses on evaluation using latency prediction, but accuracy prediction is also evaluated. Within latency prediction, both standard and adversarial tasks are considered, with strong improvements shown on the adversarial task.

**Clarity:**

Clarity of the paper could be improved significantly so that it is easier for readers outside of the specific area to understand how the method is evaluated. Many key concepts and settings are not explained, so the paper is hard to understand for someone who does not work in the exact area. E.g. what is an adversarial task in this case? It would be good to clarify that the method is not used for predicting both accuracy and latency at once using one model - assuming that is the case. There are claims of 400x speed improvement in the abstract but then this is not discussed in more depth in the results section.

Also the checklist could be filled in with information about how the items are satisfied.

Minor:
* Citations on L96 use two styles
* Typos - e.g. L47 devices.While
* Figure 3 has some text that is 90 degrees rotated and hence hard to read


**Overall Review:**

Positives:
* The paper introduces a novel search-space independent NN encoding strategy that is the first step towards search-space, task and device agnostic predictors for NAS
* The method is sample efficient and has performance competitive with current approaches, in some settings (adversarial task) leading to noticeable gains
* Evaluation is done across various types of predictors: latency and accuracy, variety of benchmarks and settings
* Extensive analyses are provided that study the method from various points of view, e.g. in terms of pre-training, training from scratch scenarios, etc.

Negatives:
* Inconsistent evaluations that make it challenging to make sound conclusions about the method - more details in the technical quality section
* The paper is difficult to read and understand - there are many settings considered, without proper explanations - more details in the clarity section
* Accuracy predictors can be of particular interest to many researchers, but the evaluation in the paper studies them in significantly less depth than latency predictors  - one would likely expect more balanced evaluation for both types of predictors


**Potential Impact On The Field Of Automl:**

The paper presents novel ideas and directions, but I find them to be presented in a rather inconsistent and hard to understand way, which may significantly limit the influence of the paper.

**Reproducibility (Optional):**

Code is provided so it should be reproducible.


**Review Confidence:**

2: You are willing to defend your assessment, but it is quite likely that you did not understand the central parts of the submission or that you are unfamiliar with some pieces of related work.

**Review Rating:**

5: Borderline Leaning Reject: Technically sound paper where reasons to reject nonetheless outweigh reasons to accept. Please use sparingly.

**Review Summary:**

The paper provides an interesting method that is a step towards more general NAS predictors. However, the paper would strongly benefit from more consistent evaluations and clearer writing so that it is easier to see the advantages of the method.


**Technical Quality And Correctness:**

The paper studies a large number of variations and settings, with approaches evaluated in different and inconsistent ways. For example in Table 1 the proposed TL approaches use 10 samples, HELP uses 20 samples and the main comparison is drawn between HELP and TL. One of the main arguments is that TL uses only 10 examples while HELP uses 20 so TL is more sample-efficient. However, both achieve very similar scores in the shown setups, so a question arises what the comparison would look like if it was consistent. Then there is Table 2 which uses 20 samples for both TL and HELP. Table 3 includes only a selection of the proposed variations, with no explanation - e.g. why is FLOPs not there? There are also claims about how the proposed method is much more sample efficient (Figure 6), but it does not show the performance of the method HELP when using fewer samples (with HELP achieving the highest score on that set-up). Pre-training of HELP could be done using fewer measurements so that a more insightful evaluation can be provided.

---

> ### Author Response · Authors · 2023-05-02
> **Response to Reviewer ur9J**
>
>
> We appreciate the reviewer's detailed feedback and constructive criticism. We would like to address the points raised and provide clarification to improve the understanding of our paper.
>
> >  there is Table 2 which uses 20 samples for both TL and HELP.
>
> We have updated the main text to explain this point further [Line 236]. Regarding Table 1, our intention was to compare our method with the baselines reported in the HELP paper. The default device sets exhibit low sensitivity to the change in the number of samples, which is why we introduced an 'adversarial' device set. Table 2 tests both our TL and HELP methods on a difficult task, showcasing the effectiveness of using hardware latency tables in conjunction with the proposed ZCP and HWL representations. Furthermore, we included Figure 10 in the Appendix to demonstrate the sample efficiency of FSP (TL) and HELP on the adversarial device sets. We chose to report only the most important results in the paper to emphasize the effectiveness of our representation.
>
> > ‘...  what is an adversarial task in this case? …’
>
> With the updated page limit, we have explained the adversarial task in the main text [Line 225, 236] with further details in the Appendix. We have also described the search spaces, hardware latency benchmarks, devices, zero cost proxies in the Appendix to aid readers outside of the area to understand the paper.
>
> > ‘... It would be good to clarify that the method is not used for predicting both accuracy and latency at once using one model - assuming that is the case.  …’
>
> Figure 3 indeed depicts different predictors and visually conveys the intended information. We have also added a sentence to clarify this in the Method. [Line 108]
>
> > ‘... in significantly less depth than latency predictors - one would likely expect more balanced evaluation for both types of predictors … ‘
>
> To evaluate the accuracy predictor in more depth, we have also added ‘NAS search’ result [Line 315] specifically focusing on ZCP vs Vec. We employed the same evaluation method for both accuracy and latency predictors. For hardware latency, we used a hardware embedding table. Our experiments are robust, with confusion matrices showcasing the improvement across a large number of search spaces. In addition, Figures 16 and 17 in the Supplemental material demonstrate substantial cross-transfer results.
>
> > ‘...checklist and FLOPs for Table 3…’
>
> We have updated the checklist. Unfortunately, we are unable to initialize individual networks on the FBNet search space, as the official code-base does not support it. Further, they have not provided the FLOPs for their neural networks. Which is why neither the reference paper (HELP) or ours include FLOPs.
>
> We acknowledge the reviewer's concerns and have updated the checklist accordingly. We have also included HELP results for Figure 6 in the Appendix to provide a more complete and insightful evaluation. We appreciate the reviewer's valuable input and suggestions. We believe that our work is an important step towards more general NAS predictors and will strive to improve the clarity and consistency of our evaluations in the revised version of the paper. In light of our clarification, we would appreciate it if the reviewer could kindly consider revising their score.

---

> > ### Author Response · Authors · 2023-05-08
> > **Response To Reviewer**
> >
> > Dear Reviewer,
> >
> > As we approach the end of the author-reviewer discussion period, we kindly request your attention to review our revised manuscript and the responses to your comments as soon as possible. This will enable us to address any further queries or concerns you may have before the discussion period concludes.
> >
> > We have made several improvements to our manuscript to incorporate the valuable feedback provided by the reviewers:
> > 1. Expanded our literature survey with relevant citations.
> > 2. Made several clarifications regarding the methodology used as well as explained Sample Hardware Embedding in greater depth.
> > 3. Employed a highly-cited methodology for neural architecture search (BRP-NAS) to evaluate the effectiveness of our ZCP based accuracy predictors (Figure 10), and included a NAS Search Effectiveness sub-section detailing our search methodology and presenting specific NAS results.
> >
> > If our rebuttal satisfactorily addresses your concerns, we kindly request that you consider revising your rating of our submission accordingly.

---

> > ### Comment · Reviewer_ur9J · 2023-05-08
> > **Response**
> >
> > Thanks for the additional clarifications and analyses. I increased my rating by one level to reflect the improvements to the paper. However, I still find that the paper suffers from inconsistent evaluations and relatively confusing explanations, even if there have been improvements to it (the response does not address the inconsistencies). Accuracy predictors are studied in more depth now, but there is still quite significant disparity between how much Hardware latency predictors and Accuracy predictors are studied.

---

### Official Review · Reviewer_WcoC · 2023-04-17

**Potential Impact On The Field Of Automl Rating:** 4
**Technical Quality And Correctness Rating:** 4
**Clarity:** I really enjoy reading this paper. It…
**Clarity Rating:** 4
**Actions Required To Increase Overall Recommendation:** I suggest the conference accept this …

**Summary Of Contributions:**

This paper proposed Multi-Predict NAS algorithm that can be used for multiple search spaces and multiple hardware devices. Multi-Predict uses the search space agnostic architecture encoding, ZCP. ZCP is an ensemble of multiple zero-cost proxies so that ZCP is independent of the search space but still can encode architecture information. Multi-Predict applies learnable embeddings to represent different devices, named HWL. Thanks to the novel encoding of architectures and hardware, Multi-Predict can train one predictor for multiple search spaces and different hardware devices. Furthermore, the encoding algorithms enable Multi-Predict to transfer knowledge to new search spaces or add new hardware devices. It is demonstrated that Multi-Predict can transfer knowledge from FBNet to NAS-Bench-201.

**Overall Review:**

This paper focuses on an important NAS topic, transferring knowledge between different search spaces. A very recent work [R1] uses meta-learning to solve this problem and transfers knowledge between different datasets. Multi-Predict is able to train a predictor for multiple search spaces and different devices. Multi-Predict proposes a very simple yet effective architecture encoding (ZCP) that is agnostic to search spaces. I really like the idea of ZCP. I suggest the authors discuss [R1] in the literature review because [R1] is the most recent paper.

This paper definitely should be accepted.

[R1] Transfer NAS with Meta-learned Bayesian Surrogates, ICLR 2023. https://openreview.net/forum?id=paGvsrl4Ntr

**Potential Impact On The Field Of Automl:**

Transferring knowledge between different search spaces is very important and promising. I expect a lot of discussion in this paper.

**Review Confidence:**

4: You are confident in your assessment, but not absolutely certain. It is unlikely, but not impossible, that you did not understand some parts of the submission or that you are unfamiliar with some pieces of related work.

**Review Rating:**

8: Accept: Technically sound paper with major impact and strong evaluation, with perhaps some minor flaws.

**Review Summary:**

Multi-Predict makes it possible to transfer knowledge between different search spaces.  ZCP encoding is simple, smart, and effective. Although ZCP does not have a very strong algorithm contribution, I really like the idea. In my opinion, ZCP is an ensemble of multiple zero-cost proxies. In that case, it can represent the architecture but is not dependent to search space. Furthermore, Multi-Predict makes it easier to adapt predictors to new search spaces or new hardware.

**Technical Quality And Correctness:**

ZCP encoding is an ensemble of multiple zero-cost proxies and HWL is a learnable embedding of hardware. They are simple yet effective.

---

> ### Author Response · Authors · 2023-05-02
> **Response to Reviewer WcoC**
>
> We would like to thank the reviewer for their insightful comments. We are pleased to find that the reviewer appreciates the potential impact of our findings.
> We have included [R1] in our related works in the updated version of the paper. We have also incorporated the comments by other reviewers and added NAS search results to demonstrate the effectiveness of our representation.

---

### Comment · Program_Chairs · 2023-03-31
**Reproducibility**

Dear authors, a reviewer has indicated that this submission does not have code. Reproducibility is a crucial axis when reviewing submissions (please see our full policy under “Reproducibility” at https://2023.automl.cc/calls/forpapers/).
Please try to submit the code as soon as you can, or else your submission will have very low marks for reproducibility. For example, you can reply to this comment, using an anonymous link such as https://anon-github.automl.cc/ .

Best, PC Chairs.

---

> ### Author Response · Authors · 2023-03-31
> **Code Link**
>
> Thank you for informing us of the missing code.
>
> Please find the code here: https://anon-github.automl.cc/r/automl_conf_codesub_multipredict-4628
>
> We have updated the README.md of the repository with an anonymized link to the data-set folder.

---

> > ### Comment · Program_Chairs · 2023-04-03
> > **Thanks**
> >
> > Thank you!

---

### Author Response · Authors · 2023-05-03
**Author Rebuttal Discussion Reminder**

Dear Reviewers,

As we are now a few days into the author-reviewer discussion period, we would greatly appreciate it if you could take the time to review our updated draft and responses to your comments at your earliest convenience. Doing so will allow us to address any additional questions or concerns you may have before the discussion period comes to a close.

If our rebuttal satisfactorily addresses your concerns, we kindly request that you consider revising your rating of our submission accordingly.

We truly value your time and effort in this process and thank you for your valuable review.